# Improving Black-box Adversarial Attacks with a Transfer-based Prior

**Shuyu Cheng**\*, **Yinpeng Dong**\*, **Tianyu Pang, Hang Su, Jun Zhu**†

Dept. of Comp. Sci. and Tech., BNRist Center, State Key Lab for Intell. Tech. & Sys.,
Institute for AI, THBI Lab, Tsinghua University, Beijing, 100084, China

`{chengsy18, dyp17, pty17}@mails.tsinghua.edu.cn, {suhangss, dcszj}@mail.tsinghua.edu.cn`

## Abstract

We consider the black-box adversarial setting, where the adversary has to generate adversarial perturbations without access to the target models to compute gradients. Previous methods tried to approximate the gradient either by using a transfer gradient of a surrogate white-box model, or based on the query feedback. However, these methods often suffer from low attack success rates or poor query efficiency since it is non-trivial to estimate the gradient in a high-dimensional space with limited information. To address these problems, we propose a prior-guided random gradient-free (P-RGF) method to improve black-box adversarial attacks, which takes the advantage of a transfer-based prior and the query information simultaneously. The transfer-based prior given by the gradient of a surrogate model is appropriately integrated into our algorithm by an optimal coefficient derived by a theoretical analysis. Extensive experiments demonstrate that our method requires much fewer queries to attack black-box models with higher success rates compared with the alternative state-of-the-art methods.

## 1 Introduction

Although deep neural networks (DNNs) have achieved significant success on various tasks [12], they have been shown to be vulnerable to adversarial examples [2, 33, 13], which are crafted to fool the models by modifying normal examples with human imperceptible perturbations. Many efforts have been devoted to studying the generation of adversarial examples, which is crucial to identify the weaknesses of deep learning algorithms [33, 1], serve as a surrogate to evaluate robustness [5], and consequently contribute to the design of robust deep learning models [24].

In general, adversarial attacks can be categorized into white-box attacks and black-box attacks. In the white-box setting, the adversary has full access to the model, and can use various gradient-based methods [13, 20, 5, 24] to generate adversarial examples. In the more challenging black-box setting, the adversary has no or limited knowledge about the model, and crafts adversarial examples without any gradient information. The black-box setting is more practical in many real-world situations.

Many methods [30, 6, 3, 7, 18, 27, 35, 19, 9] have been proposed to perform black-box adversarial attacks. A common idea is to use an approximate gradient instead of the true gradient for crafting adversarial examples. The approximate gradient could be either the gradient of a surrogate model (termed as *transfer-based* attacks) or numerically estimated by the zeroth-order optimization methods (termed as *query-based* attacks). In transfer-based attacks, adversarial examples generated for a different model are probable to remain adversarial for the target model due to the transferability [29]. Although various methods [7, 23, 8] have been introduced to improve the transferability, the attack success rate is still unsatisfactory. The reason is that there lacks an adjustment procedure in transfer-based attacks when the gradient of the surrogate model points to a non-adversarial region of the

target model. In query-based adversarial attacks, the gradient can be estimated by various methods, such as finite difference [6, 27], random gradient estimation [35], and natural evolution strategy [18]. These methods usually result in a higher attack success rate compared with the transfer-based attack methods [6, 27], but they require a tremendous number of queries to perform a successful attack. The inefficiency mainly comes from the underutilization of priors, since the current methods are nearly optimal to estimate the gradient [19].

To address the aforementioned problems and improve black-box attacks, we propose a **prior-guided random gradient-free (P-RGF)** method to utilize the transfer-based prior for query-efficient black-box attacks under the gradient estimation framework. The transfer-based prior is given by the gradient of a surrogate white-box model, which contains abundant prior knowledge of the true gradient. Our method provides a gradient estimate by querying the target model with random samples that are biased towards the transfer gradient and acquiring the corresponding loss values. We provide a theoretical analysis on deriving the optimal coefficient, which controls the strength of the transfer gradient. Our method is also flexible to integrate other forms of prior information. As a concrete example, we incorporate the commonly used *data-dependent prior* [19] into our algorithm along with the transfer-based prior. Extensive experiments demonstrate that our method significantly outperforms the previous state-of-the-art methods in terms of black-box attack success rate and query efficiency, which verifies the superiority of our method for black-box adversarial attacks.

## 2 Background

In this section, we review the background and the related work on black-box adversarial attacks.

### 2.1 Adversarial setup

Given a classifier $C(x)$ and an input-label pair $(x, y)$, the goal of attacks is to generate an adversarial example $x^{adv}$ that is misclassified while the distance between the adversarial input and the normal input measured by the $\ell_p$ norm is smaller than a preset threshold $\epsilon$ as

$$C(x^{adv}) \neq y, \text{ s.t. } \|x^{adv} - x\|_p \leq \epsilon. \tag{1}$$

Note that this corresponds to the untargeted attack. We present our framework and algorithm based on the untargeted attack for clarity, while the extension to the targeted one is straightforward.

An adversarial example can be generated by solving the constrained optimization problem as

$$x^{adv} = \underset{x':\|x'-x\|_p \leq \epsilon}{\arg\max} f(x', y), \tag{2}$$

where $f$ is a loss function on top of the classifier $C(x)$, e.g., the cross-entropy loss. Many gradient-based methods [13, 20, 5, 24] have been proposed to solve this optimization problem. The state-of-the-art projected gradient descent (PGD) [24] iteratively generates adversarial examples as

$$x^{adv}_{t+1} = \Pi_{\mathcal{B}_p(x,\epsilon)}(x^{adv}_t + \eta \cdot g_t), \tag{3}$$

where $\Pi$ is the projection operation, $\mathcal{B}_p(x, \epsilon)$ is the $\ell_p$ ball centered at $x$ with radius $\epsilon$, $\eta$ is the step size, and $g_t$ is the normalized gradient under the $\ell_p$ norm, e.g., $g_t = \frac{\nabla_x f(x^{adv}_t, y)}{\|\nabla_x f(x^{adv}_t, y)\|_2}$ under the $\ell_2$ norm, and $g_t = \text{sign}(\nabla_x f(x^{adv}_t, y))$ under the $\ell_\infty$ norm. This method requires full access to the gradient of the target model, which is designed under the white-box attack setting.

### 2.2 Black-box attacks

The direct access to the model gradient is unrealistic in many real-world applications, where we need to perform attacks in the black-box manner. We can still adopt the PGD method to generate adversarial examples, except that the true gradient $\nabla_x f(x, y)$ is usually replaced by an approximate gradient. Black-box attacks can be roughly divided into transfer-based attacks and query-based attacks. Transfer-based attacks adopt the gradient of a surrogate white-box model to generate adversarial examples, which are probable to fool the black-box model due to the transferability [30, 23, 7]. Query-based attacks estimate the gradient by the zeroth-order optimization methods, when the loss values could be accessed through queries. Chen et al. [6] propose to use the symmetric difference quotient [21] to estimate the gradient at each coordinate as

$$\hat{g}_i = \frac{f(x + \sigma e_i, y) - f(x - \sigma e_i, y)}{2\sigma} \approx \frac{\partial f(x, y)}{\partial x_i}, \tag{4}$$

where $\sigma$ is a small constant, and $e_i$ is the $i$-th unit basis vector. Although query-efficient mechanisms have been developed [6, 27], the coordinate-wise gradient estimation inherently results in the query complexity being proportional to the input dimension $D$, which is prohibitively large with high-dimensional input space, e.g., $D \approx 270,000$ for ImageNet [31]. To improve query efficiency, the approximated gradient $\hat{g}$ can be estimated by the random gradient-free (RGF) method [26, 11, 10] as

$$\hat{g} = \frac{1}{q} \sum_{i=1}^{q} \hat{g}_i, \text{ where } \hat{g}_i = \frac{f(x + \sigma u_i, y) - f(x, y)}{\sigma} \cdot u_i, \tag{5}$$

where $\{u_i\}_{i=1}^{q}$ are the random vectors independently sampled from a distribution $\mathcal{P}$ on $\mathbb{R}^D$, and $\sigma$ is the parameter to control the sampling variance. It is noted that $\hat{g}_i \to u_i^\top \nabla_x f(x, y) \cdot u_i$ when $\sigma \to 0$, which is nearly an unbiased estimator of the gradient [10] when $\mathbb{E}[u_i u_i^\top] = \mathbf{I}$. $\hat{g}$ is the average estimation over $q$ random directions to reduce the variance. The natural evolution strategy (NES) [18] is another variant of Eq. (5), which conducts the antithetic sampling over a Gaussian distribution. Ilyas et al. [19] show that these methods are nearly optimal to estimate the gradient, but their query efficiency could be improved by incorporating informative priors. They identify the time and data-dependent priors for black-box attacks. Different from the alternative methods, our proposed transfer-based prior is more effective as shown in the experiments. Moreover, the transfer-based prior can also be used together with other priors. We demonstrate the flexibility of our algorithm by incorporating the commonly used data-dependent prior as an example.

### 2.3 Black-box attacks based on both transferability and queries

There are also several works that adopt both the transferability of adversarial examples and the model queries for black-box attacks. Papernot et al. [30, 29] train a local substitute model to mimic the black-box model with a synthetic dataset, in which the labels are given by the black-box model through queries. Then the black-box model is attacked by the adversarial examples generated for the substitute model based on the transferability. A meta-model [28] can reverse-engineer the black-box model and predict its attributes (such as architecture, optimization procedure, and training data) through a sequence of queries. Given the predicted attributes of the black-box model, the attacker can find similar surrogate models, which are better to craft transferable adversarial examples against the black-box model. These methods all use queries to obtain knowledge of the black-box model, and train/find surrogate models to generate adversarial examples, with the purpose of improving the transferability. However, we do not optimize the surrogate model, but focus on utilizing the gradient of a fixed surrogate model to obtain a more accurate gradient estimate.

A recent work [4] also uses the gradient of a surrogate model to improve the efficiency of query-based black-box attacks. This method focuses on a different attack scenario, where the model only provides the hard-label outputs, but we consider the setting where the loss values could be accessed. Moreover, this method controls the strength of the transfer gradient by a preset hyperparameter, but we obtain its optimal value through a theoretical analysis based on the gradient estimation framework. It's worth mentioning that a similar but independent work [25] also uses surrogate gradients to improve zeroth-order optimization, but they did not apply their method to black-box adversarial attacks.

## 3 Methodology

In this section, we first introduce the gradient estimation framework. Then we propose the prior-guided random gradient-free (P-RGF) algorithm. We further incorporate the data-dependent prior [19] into our algorithm. We also provide an alternative algorithm for the same purpose in Appendix B.

### 3.1 Gradient estimation framework

The key challenge in black-box adversarial attacks is to estimate the gradient of a model, which can be used to conduct gradient-based attacks. In this paper, we aim to estimate the gradient $\nabla_x f(x, y)$ of the black-box model $f$ more accurately to improve black-box attacks. We denote the gradient $\nabla_x f(x, y)$ by $\nabla f(x)$ in the following for simplicity. We assume that $\nabla f(x) \neq 0$ in this paper. The objective of gradient estimation is to find the best estimator, which approximates the true gradient $\nabla f(x)$ by reaching the minimum value of the loss function as

$$\hat{g}^* = \arg\min_{\hat{g} \in \mathcal{G}} L(\hat{g}), \tag{6}$$

where $\hat{g}$ is a gradient estimator given by any estimation algorithm, $\mathcal{G}$ is the set of all possible gradient estimators, and $L(\hat{g})$ is a loss function to measure the performance of the estimator $\hat{g}$. Specifically, we let the loss function of the gradient estimator $\hat{g}$ be

$$L(\hat{g}) = \min_{b \geq 0} \mathbb{E} \|\nabla f(x) - b\hat{g}\|_2^2, \tag{7}$$

where the expectation is taken over the randomness of the estimation algorithm to obtain $\hat{g}$. The loss $L(\hat{g})$ is the minimum expected squared $\ell_2$ distance between the true gradient $\nabla f(x)$ and scaled estimator $b\hat{g}$. The previous work [35] also uses the expected squared $\ell_2$ distance $\mathbb{E}\|\nabla f(x) - \hat{g}\|_2^2$ as the loss function, which is similar to ours. However, the value of this loss function will change with different magnitude of the estimator $\hat{g}$. In generating adversarial examples, the gradient is usually normalized [13, 24], such that the direction of the gradient estimator, instead of the magnitude, will affect the performance of attacks. Thus, we incorporate a scaling factor $b$ in Eq. (7) and minimize the error w.r.t. $b$, which can neglect the impact of the magnitude on the loss of the estimator $\hat{g}$.

## 3.2 Prior-guided random gradient-free method

In this section, we present the prior-guided random gradient-free (P-RGF) method, which is a variant of the random gradient-free (RGF) method. Recall that in RGF, the gradient can be estimated via a set of random vectors $\{u_i\}_{i=1}^q$ as in Eq. (5) with $q$ being the number of random vectors. Directly using RGF without prior information will result in poor query efficiency as shown in our experiments. In our method, we propose to sample the random vectors that are biased towards the transfer gradient, to fully exploit the prior information.

Let $v$ be the normalized transfer gradient of a surrogate model such that $\|v\|_2 = 1$, and the cosine similarity between the transfer gradient and the true gradient be

$$\alpha = v^\top \overline{\nabla f(x)} \text{ with } \overline{\nabla f(x)} = \|\nabla f(x)\|_2^{-1} \nabla f(x), \tag{8}$$

where $\overline{\nabla f(x)}$ is the $\ell_2$ normalization of the true gradient $\nabla f(x)$.[1] We assume that $\alpha \geq 0$ without loss of generality, since we can reassign $v \leftarrow -v$ when $\alpha < 0$. Although the true value of $\alpha$ is unknown, we could estimate it efficiently, which will be introduced in Sec. 3.3.

For the RGF estimator $\hat{g}$ in Eq. (5), we further assume that the sampling distribution $\mathcal{P}$ is defined on the unit hypersphere in the $D$-dimensional space, such that the random vectors $\{u_i\}_{i=1}^q$ drawn from $\mathcal{P}$ satisfy $\|u_i\|_2 = 1$. Then, we can represent the loss of the RGF estimator by the following theorem.

**Theorem 1.** *(Proof in Appendix A.1) If $f$ is differentiable at $x$, the loss of the RGF estimator $\hat{g}$ is*

$$\lim_{\sigma \to 0} L(\hat{g}) = \|\nabla f(x)\|_2^2 - \frac{\left(\nabla f(x)^\top \mathbf{C} \nabla f(x)\right)^2}{(1 - \frac{1}{q})\nabla f(x)^\top \mathbf{C}^2 \nabla f(x) + \frac{1}{q}\nabla f(x)^\top \mathbf{C} \nabla f(x)}, \tag{9}$$

*where $\sigma$ is the sampling variance, $\mathbf{C} = \mathbb{E}[u_i u_i^\top]$ with $u_i$ being the random vector, $\|u_i\|_2 = 1$, and $q$ is the number of random vectors as in Eq. (5).*

Given the definition of $\mathbf{C}$, it needs to satisfy two constraints: (1) it should be positive semi-definite; (2) its trace should be 1 since $\text{Tr}(\mathbf{C}) = \mathbb{E}[\text{Tr}(u_i u_i^\top)] = \mathbb{E}[u_i^\top u_i] = 1$. It is noted from Theorem 1 that we can minimize $L(\hat{g})$ by optimizing $\mathbf{C}$, i.e., we can achieve an optimal gradient estimator by carefully sampling the random vectors $u_i$, yielding an query-efficient adversarial attack.

Specifically, $\mathbf{C}$ can be decomposed as $\sum_{i=1}^D \lambda_i v_i v_i^\top$, where $\{\lambda_i\}_{i=1}^D$ and $\{v_i\}_{i=1}^D$ are the eigenvalues and orthonormal eigenvectors of $\mathbf{C}$, and $\sum_{i=1}^D \lambda_i = 1$. In our method, since we propose to bias $u_i$ towards $v$ to exploit its prior information, we can specify an eigenvector to be $v$, and let the corresponding eigenvalue be a tunable coefficient. For the other eigenvalues, we set them to be equal since we do not have any prior knowledge about the other eigenvectors. In this case, we let

$$\mathbf{C} = \lambda v v^\top + \frac{1 - \lambda}{D - 1}(\mathbf{I} - v v^\top), \tag{10}$$

where $\lambda \in [0, 1]$ controls the strength of the transfer gradient that the random vectors $\{u_i\}_{i=1}^q$ are biased towards. We can easily construct a random vector with unit length while satisfying Eq. (10) (proof in Appendix A.2) as

$$u_i = \sqrt{\lambda} \cdot v + \sqrt{1 - \lambda} \cdot \overline{(\mathbf{I} - v v^\top)\xi_i}, \tag{11}$$

**Algorithm 1** Prior-guided random gradient-free (P-RGF) method

---

**Input:** The black-box model $f$; input $x$ and label $y$; the normalized transfer gradient $v$; sampling variance $\sigma$; number of queries $q$; input dimension $D$.
**Output:** Estimate of the gradient $\nabla f(x)$.

1: Estimate the cosine similarity $\alpha = v^\top \overline{\nabla f(x)}$ (detailed in Sec. 3.3);
2: Calculate $\lambda^*$ according to Eq. (12) given $\alpha$, $q$, and $D$;
3: **if** $\lambda^* = 1$ **then**
4:     **return** $v$;
5: **end if**
6: $\hat{g} \leftarrow \mathbf{0}$;
7: **for** $i = 1$ to $q$ **do**
8:     Sample $\xi_i$ from the uniform distribution on the $D$-dimensional unit hypersphere;
9:     $u_i = \sqrt{\lambda^*} \cdot v + \sqrt{1 - \lambda^*} \cdot (\mathbf{I} - vv^\top)\xi_i$;
10:     $\hat{g} \leftarrow \hat{g} + \dfrac{f(x + \sigma u_i, y) - f(x, y)}{\sigma} \cdot u_i$;
11: **end for**
12: **return** $\nabla f(x) \leftarrow \dfrac{1}{q}\hat{g}$.

---

where $\xi_i$ is sampled uniformly from the unit hypersphere. Hereby, the problem turns to optimizing $\lambda$ that minimizes $L(\hat{g})$. The previous work [35] can also be categorized as a special case of our method when $\lambda = \frac{1}{D}$ and $\mathbf{C} = \frac{1}{D}\mathbf{I}$, such that the random vectors are drawn from the uniform distribution on the hypersphere. When $\lambda \in [0, \frac{1}{D})$, it indicates that the transfer gradient is worse than a random vector, so we are encouraged to search in other directions by using a small $\lambda$. To find the optimal $\lambda$, we plug Eq. (10) into Eq. (9), and obtain the closed-form solution (proof in Appendix A.3) as

$$
\lambda^* = \begin{cases} 0 & \text{if } \alpha^2 \leq \dfrac{1}{D + 2q - 2} \\ \dfrac{(1 - \alpha^2)(\alpha^2(D + 2q - 2) - 1)}{2\alpha^2 Dq - \alpha^4 D(D + 2q - 2) - 1} & \text{if } \dfrac{1}{D + 2q - 2} < \alpha^2 < \dfrac{2q - 1}{D + 2q - 2} \\ 1 & \text{if } \alpha^2 \geq \dfrac{2q - 1}{D + 2q - 2} \end{cases} \quad . \quad (12)
$$

**Remark.** It can be proven (in Appendix A.4) that $\lambda^*$ is a monotonically increasing function of $\alpha^2$, and a monotonically decreasing function of $q$ (when $\alpha^2 > \frac{1}{D}$). It means that a larger $\alpha$ or a smaller $q$ (when the transfer gradient is not worse than a random vector) would result in a larger $\lambda^*$, which makes sense since we tend to rely on the transfer gradient more when (1) it approximates the true gradient better; (2) the number of queries is not enough to provide much gradient information.

We summarize the P-RGF method in Algorithm 1. Note that when $\lambda^* = 1$, we directly return the transfer gradient as the estimate of $\nabla f(x)$, which can save many queries.

### 3.3 Estimation of cosine similarity

To complete our algorithm, we also need to estimate $\alpha = v^\top \overline{\nabla f(x)} = \frac{v^\top \nabla f(x)}{\|\nabla f(x)\|_2}$, where $v$ is the normalized transfer gradient. Note that the inner product $v^\top \nabla f(x)$ can be easily estimated by the finite difference method

$$
v^\top \nabla f(x) \approx \frac{f(x + \sigma v, y) - f(x, y)}{\sigma}, \quad (13)
$$

using a small $\sigma$. Hence, the problem is reduced to estimating $\|\nabla f(x)\|_2$.

Suppose that it is allowed to conduct $S$ queries to estimate $\|\nabla f(x)\|_2$. We first draw a different set of $S$ random vectors $\{w_s\}_{s=1}^S$ independently and uniformly from the $D$-dimensional unit hypersphere, and then estimate $w_s^\top \nabla f(x)$ using Eq. (13). Suppose that we have a $r$-degree homogeneous function $g$ of $S$ variables, i.e., $g(az) = a^r g(z)$ where $a \in \mathbb{R}$ and $z \in \mathbb{R}^S$, then we have

$$
g(\mathbf{W}^\top \nabla f(x)) = \|\nabla f(x)\|_2^r \cdot g(\mathbf{W}^\top \overline{\nabla f(x)}), \quad (14)
$$

where $\mathbf{W}$ is the collection of the random vectors as $\mathbf{W} = [w_1, ..., w_S]$. In this case, the norm of the gradient $\|\nabla f(x)\|_2$ could be computed easily if both $g(\mathbf{W}^\top \nabla f(x))$ and $g(\mathbf{W}^\top \overline{\nabla f(x)})$ are available. Note that $g(\mathbf{W}^\top \nabla f(x))$ can be calculated since each $w_s^\top \nabla f(x)$ is available.

However, it is non-trivial to obtain the value of $w_s^\top \overline{\nabla f(x)}$ as well as the function value $g\big(\mathbf{W}^\top \overline{\nabla f(x)}\big)$. Nevertheless, we note that the distribution of $w_s^\top \overline{\nabla f(x)}$ is the same regardless of the direction of $\overline{\nabla f(x)}$, thus we can compute the expectation of the function value $\mathbb{E}\big[g\big(\mathbf{W}^\top \overline{\nabla f(x)}\big)\big]$. Based on that, we use $\frac{g(\mathbf{W}^\top \overline{\nabla f(x)})}{\mathbb{E}[g(\mathbf{W}^\top \overline{\nabla f(x)})]}$ as an unbiased estimator of $\|\nabla f(x)\|_2^r$. In particular, we choose $g$ as $g(z) = \frac{1}{S}\sum_{s=1}^{S} z_s^2$. Then $r = 2$, and we have

$$\mathbb{E}\big[g\big(\mathbf{W}^\top \overline{\nabla f(x)}\big)\big] = \mathbb{E}\big[(w_1^\top \overline{\nabla f(x)})^2\big] = \overline{\nabla f(x)}^\top \mathbb{E}[w_1 w_1^\top]\overline{\nabla f(x)} = \frac{1}{D}. \tag{15}$$

By plugging Eq. (15) into Eq. (14), we can estimate the gradient norm by

$$\|\nabla f(x)\|_2 \approx \sqrt{\frac{D}{S}\sum_{s=1}^{S}(w_s^\top \nabla f(x))^2} \approx \sqrt{\frac{D}{S}\sum_{s=1}^{S}\Big(\frac{f(x+\sigma w_s, y) - f(x, y)}{\sigma}\Big)^2}. \tag{16}$$

To save queries, we estimate the gradient norm periodically instead of in every iteration, since usually it does not change very fast in the optimization process.

### 3.4 Incorporating the data-dependent prior

The proposed P-RGF method is generally flexible to integrate other priors. As a concrete example, we incorporate the commonly used data-dependent prior [19] along with the transfer-based prior into our algorithm. The data-dependent prior is proposed to reduce query complexity, which suggests that we can utilize the structure of the inputs to reduce the input-space dimension without sacrificing much estimation accuracy. This idea has also been adopted in several works [6, 35, 14, 4]. We observe that many works restrict the adversarial perturbations to lie in a linear subspace of the input space, which allows the application of our theoretical framework.

Consider the RGF estimator in Eq. (5). To leverage the data-dependent prior, suppose $u_i = \mathbf{V}\xi_i$, where $\mathbf{V} = [v_1, v_2, ..., v_d]$ is a $D \times d$ matrix ($d < D$), $\{v_j\}_{j=1}^{d}$ is an orthonormal basis in the $d$-dimensional subspace of the input space, and $\xi_i$ is a random vector sampled from the $d$-dimensional unit hypersphere. When $\xi_i$ is sampled from the uniform distribution, $\mathbf{C} = \frac{1}{d}\sum_{i=1}^{d} v_i v_i^\top$.

Specifically, we focus on the data-dependent prior in [19]. In this method, the random vector $\xi_i$ drawn in $\mathbb{R}^d$ is up-sampled to $u_i$ in $\mathbb{R}^D$ by the nearest neighbor algorithm, where $d < D$. The orthonormal basis $\{v_j\}_{j=1}^{d}$ can be obtained by first up-sampling the standard basis in $\mathbb{R}^d$ with the same method and then applying normalization.

Now we consider incorporating the data-dependent prior into our algorithm. Similar to Eq. (10), we let one eigenvector of $\mathbf{C}$ be $v$ to exploit the transfer-based prior, and the others are given by the orthonormal basis in the subspace to exploit the data-dependent prior, as

$$\mathbf{C} = \lambda v v^\top + \frac{1-\lambda}{d}\sum_{i=1}^{d} v_i v_i^\top. \tag{17}$$

By plugging Eq. (17) into Eq. (9), we can also obtain the optimal $\lambda$ (proof in Appendix A.5) as

$$\lambda^* = \begin{cases} 0 & \text{if } \alpha^2 \leq \dfrac{A^2}{d+2q-2} \\[2mm] \dfrac{A^2(A^2 - \alpha^2(d+2q-2))}{A^4 + \alpha^4 d^2 - 2A^2\alpha^2(q+dq-1)} & \text{if } \dfrac{A^2}{d+2q-2} < \alpha^2 < \dfrac{A^2(2q-1)}{d} \\[2mm] 1 & \text{if } \alpha^2 \geq \dfrac{A^2(2q-1)}{d} \end{cases}, \tag{18}$$

where $A^2 = \sum_{i=1}^{d}(v_i^\top \overline{\nabla f(x)})^2$. Note that $A$ should also be estimated. We use a similar method to the one for estimating $\alpha$ in Sec. 3.3, which is provided in Appendix C.

The remaining problem is to construct a random vector $u_i$ satisfying $\mathbb{E}[u_i u_i^\top] = \mathbf{C}$, with $\mathbf{C}$ defined in Eq. (17). In general, it is difficult since $v$ is not orthogonal to the subspace. To address this issue, we sample $u_i$ in a way that $\mathbb{E}[u_i u_i^\top]$ is a good approximation of $\mathbf{C}$ (explanation in Appendix A.6), which is similar to Eq. (11) as

$$u_i = \sqrt{\lambda} \cdot v + \sqrt{1-\lambda} \cdot \overline{(\mathbf{I} - v v^\top)\mathbf{V}\xi_i}, \tag{19}$$

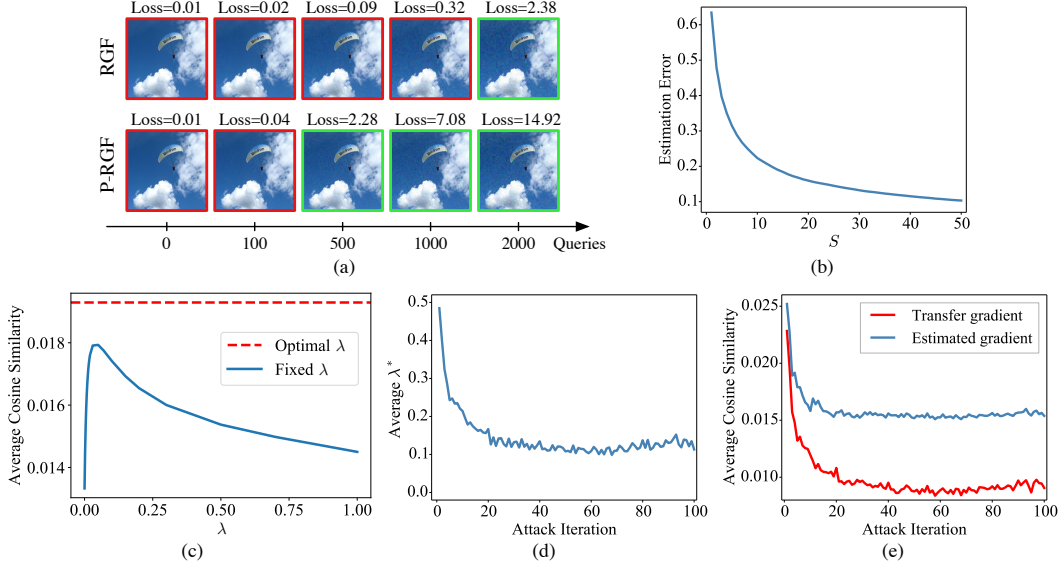

Figure 1: (a) The crafted adversarial examples for the Inception-v3 [34] model by RGF and our P-RGF w.r.t. number of queries. We show the cross-entropy loss of each image. The images in the green boxes are successfully misclassified, while those in the red boxes are not. (b) The estimation error of gradient norm with different $S$. (c) The average cosine similarity between the estimated gradient and the true gradient. The estimate is given by our method with fixed $\lambda$ and optimal $\lambda$, respectively. (d) The average $\lambda^*$ across attack iterations. (e) The average cosine similarity between the transfer and the true gradients, and that between the estimated and the true gradients, across attack iterations.

where $\xi_i$ is sampled uniformly from the $d$-dimensional unit hypersphere.

Our algorithm with the data-dependent prior is similar to Algorithm 1. We first estimate $\alpha$ and $A$, and then calculate $\lambda^*$ by Eq. (18). If $\lambda^* = 1$, we use the transfer gradient $v$ as the estimate. If not, we sample $q$ random vectors by Eq. (19) and obtain the gradient estimation by Eq. (5).

## 4 Experiments

In this section, we present the experimental results to demonstrate the effectiveness of the proposed method on attacking black-box classifiers.[2] We perform untargeted attacks under both the $\ell_2$ and $\ell_\infty$ norms on the ImageNet dataset [31]. We choose 1,000 images randomly from the validation set for evaluation. Due to the space limitation, we leave the results based on the $\ell_\infty$ norm in Appendix D. The results for both norms are consistent. For all experiments, we use the ResNet-152 model [17] as the surrogate model to generate the transfer gradient. We apply the PGD algorithm [24] to generate adversarial examples with the estimated gradient given by each method. We set the perturbation size as $\epsilon = \sqrt{0.001 \cdot D}$ and the learning rate as $\eta = 2$ in PGD under the $\ell_2$ norm, with images in $[0, 1]$.

### 4.1 Performance of gradient estimation

In this section, we conduct several experiments to show the performance of gradient estimation. All experiments in this section are performed on the Inception-v3 [34] model.

First, we show the performance of gradient norm estimation in Sec. 3.3. The gradient norm (or cosine similarity) is easier to estimate than the true gradient since it's a scalar value. Fig. 1(b) shows the estimation error of the gradient norm, defined as the (normalized) RMSE—$\sqrt{\mathbb{E}\left(\frac{\|\widehat{\nabla f(x)}\|_2 - \|\nabla f(x)\|_2}{\|\nabla f(x)\|_2}\right)^2}$, w.r.t. the number of queries $S$, where $\|\nabla f(x)\|_2$ is the true norm and $\|\widehat{\nabla f(x)}\|_2$ is the estimated one. We choose $S = 10$ in all experiments to reduce the number of queries while the estimation error is

Table 1: The experimental results of black-box attacks against Inception-v3, VGG-16, and ResNet-50 under the $\ell_2$ norm. We report the attack success rate (ASR) and the average number of queries (AVG. Q) needed to generate an adversarial example over successful attacks.

| Methods | Inception-v3 | | VGG-16 | | ResNet-50 | |
|---|---|---|---|---|---|---|
| | ASR | AVG. Q | ASR | AVG. Q | ASR | AVG. Q |
| NES [18] | 95.5% | 1718 | 98.7% | 1081 | 98.4% | 969 |
| Bandits$_T$ [19] | 92.4% | 1560 | 94.0% | 584 | 96.2% | 1076 |
| Bandits$_{TD}$ [19] | 97.2% | 874 | 94.9% | 278 | 96.8% | 512 |
| AutoZoom [35] | 85.4% | 2443 | 96.2% | 1589 | 94.8% | 2065 |
| RGF | 97.7% | 1309 | **99.8%** | 935 | 99.5% | 809 |
| P-RGF ($\lambda = 0.5$) | 96.5% | 1119 | 97.3% | 1075 | 98.3% | 990 |
| P-RGF ($\lambda = 0.05$) | 97.8% | 1021 | 99.7% | 888 | **99.6%** | 790 |
| P-RGF ($\lambda^*$) | **98.1%** | **745** | **99.8%** | **521** | **99.6%** | **452** |
| RGF$_D$ | **99.1%** | 910 | **100.0%** | 464 | **99.8%** | 521 |
| P-RGF$_D$ ($\lambda = 0.5$) | 98.2% | 1047 | 99.3% | 917 | 99.3% | 893 |
| P-RGF$_D$ ($\lambda = 0.05$) | **99.1%** | 754 | 99.9% | 482 | **99.6%** | 526 |
| P-RGF$_D$ ($\lambda^*$) | **99.1%** | **649** | 99.7% | **370** | **99.6%** | **352** |

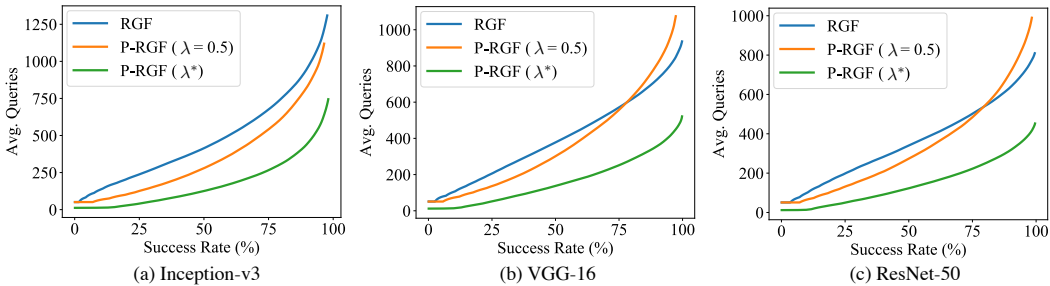

| (a) Inception-v3 | (b) VGG-16 | (c) ResNet-50 |
|---|---|---|

Figure 2: We show the average number of queries per successful image at any desired success rate.

acceptable. We also estimate the gradient norm every 10 attack iterations in all experiments to reduce the number of queries, since usually its value is relatively stable in the optimization process.

Second, we verify the effectiveness of the derived optimal $\lambda$ (i.e., $\lambda^*$) in Eq. (12) for gradient estimation, compared with any fixed $\lambda \in [0, 1]$. We perform attacks against Inception-v3 using P-RGF with $\lambda^*$, and calculate the cosine similarity between the estimated gradient and the true gradient. We calculate $\lambda^*$ using the estimated $\alpha$ in Sec. 3.3 instead of its true value. Meanwhile, along the PGD updates, we also use fixed $\lambda$ to get gradient estimates, and calculate the corresponding cosine similarities. Note that $\lambda^*$ does not correspond to any fixed value, since it varies during iterations.

The average cosine similarities of different values of $\lambda$ are shown in Fig. 1(c). It can be observed that when a suitable value of $\lambda$ is chosen, the proposed P-RGF provides a better gradient estimate than both the original RGF method with uniform distribution (when $\lambda = \frac{1}{D} \approx 0$) and the transfer gradient (when $\lambda = 1$). Adopting $\lambda^*$ brings further improvement upon any fixed $\lambda$, demonstrating the applicability of our theoretical framework.

Finally, we show the average $\lambda^*$ over all images w.r.t. attack iterations in Fig 1(d). It shows that $\lambda^*$ decreases along with the iterations. Fig. 1(e) shows the average cosine similarity between the transfer and the true gradients, and that between the estimated and the true gradients, across iterations. The results show that the transfer gradient is useful at beginning, and becomes less useful along with the iterations. However, the estimated gradient can remain higher cosine similarity with the true gradient, which facilitates the adversarial attacks consequently. The results also prove that we need to use the adaptive $\lambda^*$ in different attack iterations.

## 4.2 Results of black-box attacks on normal models

In this section, we perform attacks against three normally trained models, which are Inception-v3 [34], VGG-16 [32], and ResNet-50 [16]. We compare the proposed prior-guided random gradient-free (P-RGF) method with two baseline methods. The first is the original RGF method with uniform sampling. The second is the P-RGF method with a fixed $\lambda$, which is set to $0.5$ or $0.05$. In these

Table 2: The experimental results of black-box attacks against JPEG compression [15], randomization [37], and guided denoiser [22] under the $\ell_2$ norm. We report the attack success rate (ASR) and the average number of queries (AVG. Q) needed to generate an adversarial example over successful attacks.

| Methods | JPEG Compression [15] | | Randomization [37] | | Guided Denoiser [22] | |
|---|---|---|---|---|---|---|
| | ASR | AVG. Q | ASR | AVG. Q | ASR | AVG. Q |
| NES [18] | 47.3% | 3114 | 23.2% | 3632 | 48.0% | 3633 |
| SPSA [36] | 40.0% | 2744 | 9.6% | 3256 | 46.0% | 3526 |
| RGF | 41.5% | 3126 | 19.5% | 3259 | 50.3% | 3569 |
| P-RGF | **61.4%** | **2419** | **60.4%** | **2153** | **51.4%** | **2858** |
| RGF$_D$ | 70.4% | 2828 | 54.9% | 2819 | 83.7% | 2230 |
| P-RGF$_D$ | **81.1%** | **2120** | **82.3%** | **1816** | **89.6%** | **1784** |

methods, we set the number of queries as $q = 50$, and the sampling variance as $\sigma = 0.0001 \cdot \sqrt{D}$. We also incorporate the data-dependent prior into these three methods for comparison (which are denoted by adding a subscript "D"). We set the dimension of the subspace as $d = 50 \times 50 \times 3$. Besides, our method is compared with the state-of-the-art attack methods, including the natural evolution strategy (NES) [18], bandit optimization methods (Bandits$_T$ and Bandits$_{TD}$) [19], and AutoZoom [35]. For all methods, we restrict the maximum number of queries for each image to be 10,000. We report a successful attack if a method can generate an adversarial example within 10,000 queries and the size of perturbation is smaller than the budget (i.e., $\epsilon = \sqrt{0.001 \cdot D}$).

We show the success rate of black-box attacks and the average number of queries needed to generate an adversarial example over the successful attacks in Table 1. It can be seen that our method generally leads to higher attack success rates and requires much fewer queries than other methods. Using a fixed $\lambda$ cannot give a satisfactory result, which demonstrates the necessity of using the optimal $\lambda$ in our method. The results also show that the data-dependent prior is orthogonal to the proposed transfer-based prior, since integrating the data-dependent prior leads to better results. We show an example of attacks in Fig. 1(a). Fig. 2 shows the average number of queries over successful images by reaching a desired success rate. Our method is much more query-efficient than baseline methods.

### 4.3 Results of black-box attacks on defensive models

We further validate the effectiveness of the proposed method on attacking several defensive models, including JPEG compression [15], randomization [37], and guided denoiser [22]. We utilize the Inception-v3 model as the backbone classifier for the JPEG compression and randomization defenses. We compare P-RGF with RGF, NES [18], and SPSA [36]. The experimental settings are the same with those of attacking the normal models in Sec. 4.2. In our method, we use a smoothed version of the transfer gradient [8] as the transfer-based prior for black-box attacks, since the smoothed transfer gradient is better to defeat defensive models. The results in Table 2 also demonstrate the superiority of our method for attacking the defensive models. Our method leads to much higher attack success rates than other methods ($20\% \sim 40\%$ improvements in many cases), and also reduces the query complexity.

## 5 Conclusion

In this paper, we proposed a prior-guided random gradient-free (P-RGF) method to utilize the transfer-based prior for improving black-box adversarial attacks. Our method appropriately integrated the transfer gradient of a surrogate white-box model by the derived optimal coefficient. The experimental results consistently demonstrate the effectiveness of our method, which requires much fewer queries to attack black-box models with higher success rates.

## Acknowledgements

This work was supported by the National Key Research and Development Program of China (No. 2017YFA0700904), NSFC Projects (Nos. 61620106010, 61621136008, 61571261), Beijing NSF Project (No. L172037), Beijing Academy of Artificial Intelligence (BAAI), Tiangong Institute for Intelligent Computing, the JP Morgan Faculty Research Program and the NVIDIA NVAIL Program with GPU/DGX Acceleration.

## Footnotes

\*Equal contribution. †Corresponding author.

[1] We will use $\bar{e}$ to denote the $\ell_2$ normalization of a vector $e$ in this paper.

[2]Our code is available at: `https://github.com/thu-ml/Prior-Guided-RGF`.

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
