[Supplementary Material]

# Supplementary Material for: Improving Black-box Adversarial Attacks with a Transfer-based Prior

**Shuyu Cheng**[*], **Yinpeng Dong**[*], **Tianyu Pang, Hang Su, Jun Zhu**[†]
Dept. of Comp. Sci. and Tech., BNRist Center, State Key Lab for Intell. Tech. & Sys.,
Institute for AI, THBI Lab, Tsinghua University, Beijing, 100084, China
{chengsy18, dyp17, pty17}@mails.tsinghua.edu.cn, {suhangss, dcszj}@mail.tsinghua.edu.cn

## A  Proofs

We provide the proofs in this section.

### A.1  Proof of Theorem 1

**Theorem 1.** *If $f$ is differentiable at $x$, the loss of the RGF estimator $\hat{g}$ is*

$$\lim_{\sigma \to 0} L(\hat{g}) = \|\nabla f(x)\|_2^2 - \frac{\left(\nabla f(x)^\top \mathbf{C} \nabla f(x)\right)^2}{(1 - \frac{1}{q})\nabla f(x)^\top \mathbf{C}^2 \nabla f(x) + \frac{1}{q}\nabla f(x)^\top \mathbf{C} \nabla f(x)},$$

*where $\sigma$ is the sampling variance, $\mathbf{C} = \mathbb{E}[u_i u_i^\top]$ with $u_i$ being the random vector, $\|u_i\|_2 = 1$, and $q$ is the number of random vectors as in Eq. (5).*

**Remark 1.** *Rigorously speaking, we assume $\nabla f(x)^\top \mathbf{C} \nabla f(x) \neq 0$ in the statement of the theorem (and also in the proof), since when $\nabla f(x)^\top \mathbf{C} \nabla f(x) \neq 0$, both the numerator and the denominator of the fraction above are zero. When $\nabla f(x)^\top \mathbf{C} \nabla f(x) = 0$, $u_i^\top \nabla f(x) = 0$ holds almost surely, which implies that $L(\hat{g}) = \|\nabla f(x)\|^2$ regardless of the value of $\sigma$. In fact, this case will not happen almost surely. In the setting of black-box attacks, we cannot even design a $\mathbf{C}$ with trace 1 such that $\nabla f(x)^\top \mathbf{C} \nabla f(x) = 0$ since $\nabla f(x)$ is unknown.*

*Proof.* First, we derive $L(\hat{g})$ based on the assumption that the single estimate $\hat{g}_i$ in Eq. (5) is equal to $u_i^\top \nabla f(x) \cdot u_i$, which will hold when $f$ is locally linear.

**Lemma 1.** *Assume that the single estimate $\hat{g}_i$ in Eq. (5) is equal to $u_i^\top \nabla f(x) \cdot u_i$. We have*

$$L(\hat{g}) = \|\nabla f(x)\|_2^2 - \frac{(\nabla f(x)^\top \mathbf{C} \nabla f(x))^2}{(1 - \frac{1}{q})\nabla f(x)^\top \mathbf{C}^2 \nabla f(x) + \frac{1}{q}\nabla f(x)^\top \mathbf{C} \nabla f(x)}. \qquad (A.1)$$

*Proof.* First, we have

$$\mathbb{E}\|\nabla f(x) - b\hat{g}\|_2^2 = \|\nabla f(x)\|_2^2 - 2b\nabla f(x)^\top \mathbb{E}[\hat{g}] + b^2 \mathbb{E}\|\hat{g}\|_2^2.$$

We have $\nabla f(x)^\top \mathbb{E}[\hat{g}] = \nabla f(x)^\top \mathbb{E}[\hat{g}_i] = \mathbb{E}[\nabla f(x)^\top u_i u_i^\top \nabla f(x)] = \mathbb{E}[(\nabla f(x)^\top u_i)^2] \geq 0$. Hence

$$L(\hat{g}) = \min_{b \geq 0} \mathbb{E}\|\nabla f(x) - b\hat{g}\|_2^2 = \min_b \mathbb{E}\|\nabla f(x) - b\hat{g}\|_2^2 = \|\nabla f(x)\|_2^2 - \frac{(\nabla f(x)^\top \mathbb{E}[\hat{g}])^2}{\mathbb{E}\|\hat{g}\|_2^2}. \quad (A.2)$$

---

[*]Equal contribution. [†]Corresponding author.

Since $\hat{g}_i = u_i^\top \nabla f(x) \cdot u_i$, and $u_i^\top u_i \equiv 1$, we have

$$\mathbb{E}[\hat{g}_i] = \mathbf{C}\nabla f(x),$$

$$\begin{aligned}
\mathbb{E}\|\hat{g}_i\|_2^2 &= \mathbb{E}[\hat{g}_i^\top \hat{g}_i] \\
&= \mathbb{E}[\nabla f(x)^\top u_i u_i^\top u_i u_i^\top \nabla f(x)] \\
&= \nabla f(x)^\top \mathbb{E}[u_i(u_i^\top u_i)u_i^\top]\nabla f(x) \\
&= \nabla f(x)^\top \mathbb{E}[u_i u_i^\top]\nabla f(x) \\
&= \nabla f(x)^\top \mathbf{C}\nabla f(x).
\end{aligned}$$

Given $\mathbb{E}[\hat{g}_i]$ and $\mathbb{E}\|\hat{g}_i\|^2$, the corresponding moments of $\hat{g}$ can be computed as

$$\begin{aligned}
\mathbb{E}[\hat{g}] &= \mathbb{E}[\hat{g}_i] \tag{A.3} \\
&= \mathbf{C}\nabla f(x), \\
\mathbb{E}\|\hat{g}\|_2^2 &= \mathbb{E}\|\hat{g} - \mathbb{E}[\hat{g}]\|_2^2 + \|\mathbb{E}[\hat{g}]\|_2^2 \\
&= \frac{1}{q}\mathbb{E}\|\hat{g}_i - \mathbb{E}[\hat{g}_i]\|_2^2 + \|\mathbb{E}[\hat{g}_i]\|_2^2 \\
&= \frac{1}{q}\mathbb{E}\|\hat{g}_i\|_2^2 + (1 - \frac{1}{q})\|\mathbb{E}[\hat{g}_i]\|_2^2 \tag{A.4} \\
&= (1 - \frac{1}{q})\nabla f(x)^\top \mathbf{C}^2 \nabla f(x) + \frac{1}{q}\nabla f(x)^\top \mathbf{C}\nabla f(x).
\end{aligned}$$

Plug them into Eq. (A.2) and we complete the proof. $\qquad\square$

Next, we prove that if $f$ is not locally linear, as long as it is differentiable at $x$, then by picking a sufficient small $\sigma$, the loss tends to be that of the local linear approximation.

**Lemma 2.** *If $f$ is differentiable at $x$, letting $L_0$ denote the right-hand side of Eq. (A.1), then we have*

$$\lim_{\sigma \to 0} L(\hat{g}) = L_0.$$

*Proof.* Let $\hat{g}_i' = u_i^\top \nabla f(x) \cdot u_i$, $\hat{g}' = \frac{1}{q}\sum_{i=1}^q \hat{g}_i'$. Then $L_0 = L(\hat{g}')$. By Eq. (A.2), Eq. (A.3) and Eq. (A.4), it suffices to prove $\lim_{\sigma \to 0}\mathbb{E}[\hat{g}_i] = \mathbb{E}[\hat{g}_i']$ and $\lim_{\sigma \to 0}\mathbb{E}\|\hat{g}_i\|_2^2 = \mathbb{E}\|\hat{g}_i'\|_2^2$.

For clarity, we redefine the notation: We omit the subscript $i$, make the dependence of $\hat{g}_i$ on $\sigma$ explicit (let $\hat{g}_\sigma$ denote $\hat{g}_i$), and let $\hat{g}_0$ denote $\hat{g}_i'$. Then we omit the hat in $\hat{g}$. That is, let $g_0 \triangleq u^\top \nabla f(x) \cdot u$ and $g_\sigma \triangleq \frac{f(x+\sigma u) - f(x)}{\sigma} \cdot u$, where $u$ is sampled uniformly from the unit hypersphere. Then we want to prove $\lim_{\sigma \to 0}\mathbb{E}[g_\sigma] = \mathbb{E}[g_0]$ and $\lim_{\sigma \to 0}\mathbb{E}\|g_\sigma\|_2^2 = \mathbb{E}\|g_0\|_2^2$.

Since $f$ is differentiable at $x$, we have

$$\lim_{\sigma \to 0} \sup_{\|u\|_2 = 1} \left| \frac{f(x + \sigma u) - f(x)}{\sigma} - u^\top \nabla f(x) \right| = 0. \tag{A.5}$$

Since $\|u\|_2 \equiv 1$, we have

$$\lim_{\sigma \to 0} \mathbb{E}\|g_\sigma - g_0\|_2 \leq \lim_{\sigma \to 0} \sup_{\|u\|_2 = 1} \left| \frac{f(x + \sigma u) - f(x)}{\sigma} - u^\top \nabla f(x) \right| = 0,$$

$$\lim_{\sigma \to 0} \mathbb{E}\|g_\sigma - g_0\|_2^2 \leq \lim_{\sigma \to 0} \sup_{\|u\|_2 = 1} \left| \frac{f(x + \sigma u) - f(x)}{\sigma} - u^\top \nabla f(x) \right|^2 = 0.$$

Applying Jensen's inequality to convex function $\|\cdot\|_2$, we have $\|\mathbb{E}[g_\sigma] - \mathbb{E}[g_0]\|_2 \leq \mathbb{E}\|g_\sigma - g_0\|_2$. Since $\lim_{\sigma \to 0}\mathbb{E}\|g_\sigma - g_0\|_2 = 0$, we have $\lim_{\sigma \to 0}\mathbb{E}[g_\sigma] = \mathbb{E}[g_0]$.

Since $\left| \|g_\sigma\|_2 - \|g_0\|_2 \right| \leq \|g_\sigma - g_0\|_2$, $\lim_{\sigma \to 0}\mathbb{E}\|g_\sigma - g_0\|_2 = 0$ and $\lim_{\sigma \to 0}\mathbb{E}\|g_\sigma - g_0\|_2^2 = 0$, we have $\lim_{\sigma \to 0}\mathbb{E}\left| \|g_\sigma\|_2 - \|g_0\|_2 \right| = 0$ and $\lim_{\sigma \to 0}\mathbb{E}(\|g_\sigma\|_2 - \|g_0\|_2)^2 = 0$. Also, we have

$\|g_0\|_2 \leq \|\nabla f(x)\|_2$. Hence, we have

$$
\begin{aligned}
\lim_{\sigma \to 0} \left| \mathbb{E}\|g_\sigma\|_2^2 - \mathbb{E}\|g_0\|_2^2 \right| &\leq \lim_{\sigma \to 0} \mathbb{E}\left| \|g_\sigma\|_2^2 - \|g_0\|_2^2 \right| \\
&= \lim_{\sigma \to 0} \mathbb{E}\left[ \left| \|g_\sigma\|_2 - \|g_0\|_2 \right| \left( \|g_\sigma\|_2 + \|g_0\|_2 \right) \right] \\
&\leq \lim_{\sigma \to 0} \mathbb{E}\left[ \left( \|g_\sigma\|_2 - \|g_0\|_2 \right)^2 + 2\|g_0\|_2 \left| \|g_\sigma\|_2 - \|g_0\|_2 \right| \right] \\
&\leq \lim_{\sigma \to 0} \mathbb{E}\left[ \left( \|g_\sigma\|_2 - \|g_0\|_2 \right)^2 + 2\|\nabla f(x)\|_2 \left| \|g_\sigma\|_2 - \|g_0\|_2 \right| \right] \\
&= 0.
\end{aligned}
$$

The proof is complete. $\qquad\square$

By combining the two lemmas above, our proof for Theorem 1 is complete. $\qquad\square$

### A.2 Proof of Eq. (11)

Suppose $v$ is a fixed random vector and $\|v\|_2 = 1$. Let the $D$-dimensional random vector $u$ be

$$
u = \sqrt{\lambda} \cdot v + \sqrt{1 - \lambda} \cdot \overline{(\mathbf{I} - vv^\top)\xi}, \tag{A.6}
$$

where $\xi$ is sampled uniformly from the unit hypersphere. We want to prove that

$$
\mathbb{E}[uu^\top] = \lambda vv^\top + \frac{1 - \lambda}{D - 1}(\mathbf{I} - vv^\top). \tag{A.7}
$$

*Proof.* Let $r \triangleq \overline{(\mathbf{I} - vv^\top)\xi}$. We choose an orthonormal basis $\{v_1, ..., v_D\}$ of $\mathbb{R}^D$ such that $v_1 = v$. Then $\xi$ can be written as $\xi = \sum_{i=1}^{D} a_i v_i$, where $a = (a_1, ..., a_D)^\top$ is sampled uniformly from the unit hypersphere. Hence $(\mathbf{I} - vv^\top)\xi = \sum_{i=2}^{D} a_i v_i$, and $r = \frac{\sum_{i=2}^{D} a_i v_i}{\sqrt{\sum_{i=2}^{D} a_i^2}}$. Let $b_i = \frac{a_i}{\sqrt{\sum_{i=2}^{D} a_i^2}}$ for $i = 2, 3, ..., D$, then $b = (b_2, b_3, ..., b_D)^\top$ is sampled uniformly from the $(D - 1)$-dimensional unit hypersphere, and $r = \sum_{i=2}^{D} b_i v_i$. Hence $\mathbb{E}[r] = 0$. To compute $\mathbb{E}[rr^\top]$, we need a lemma first.

**Lemma 3.** *Suppose $d$ is a positive integer, $u = \sum_{i=1}^{d} a_i v_i$ where $a = (a_1, ..., a_d)^\top$ is sampled uniformly from the $d$-dimensional unit hypersphere, then $\mathbb{E}[uu^\top] = \frac{1}{d} \sum_{i=1}^{d} v_i v_i^\top$.*

*Proof.* $\mathbb{E}[uu^\top] = \mathbb{E}[(\sum_{i=1}^{d} a_i v_i)(\sum_{j=1}^{d} a_j v_j^\top)] = \sum_{i=1}^{d} \sum_{j=1}^{d} v_i v_j^\top \mathbb{E}[a_i a_j]$. By symmetry, we have $\mathbb{E}[a_i a_j] = 0$ when $i \neq j$, and $\mathbb{E}[a_i^2] = \mathbb{E}[a_j^2]$ for any $i, j$. Since $\sum_{i=1}^{d} a_i^2 = 1$, we have $\mathbb{E}[a_i^2] = \frac{1}{d}$ for any $i$. Hence $\mathbb{E}[uu^\top] = \frac{1}{d} \sum_{i=1}^{d} v_i v_i^\top$. $\qquad\square$

Using the lemma, we have $\mathbb{E}[rr^\top] = \frac{1}{D-1} \sum_{i=2}^{D} v_i v_i^\top = \frac{1}{D-1}(\mathbf{I} - vv^\top)$. Since $\mathbb{E}[r] = 0$, we have $\mathbb{E}[vr^\top] = \mathbb{E}[rv^\top] = 0$. Hence, we have

$$
\begin{aligned}
\mathbb{E}[uu^\top] &= \mathbb{E}[(\sqrt{\lambda} \cdot v + \sqrt{1 - \lambda} \cdot r)(\sqrt{\lambda} \cdot v + \sqrt{1 - \lambda} \cdot r)^\top] \\
&= \lambda vv^\top + (1 - \lambda)\mathbb{E}[rr^\top] \\
&= \lambda vv^\top + \frac{1 - \lambda}{D - 1}(\mathbf{I} - vv^\top).
\end{aligned}
$$

The proof is complete. $\qquad\square$

**Remark 2.** *The construction of the random vector $u$ such that $\mathbb{E}[uu^\top] = \lambda vv^\top + \frac{1-\lambda}{D-1}(\mathbf{I} - vv^\top)$ is not unique. One can choose a different kind of distribution or simply take the negative of $u$ while remaining $\mathbb{E}[uu^\top]$ invariant.*

### A.3 Proof of Eq. (12)

Let $\alpha = v^\top \overline{\nabla f(x)}$. Suppose $D \geq 2$, $q \geq 1$. After plugging Eq. (10) into Eq. (9), the optimal $\lambda$ is given by

$$\lambda^* = \begin{cases} 0 & \text{if } \alpha^2 \leq \dfrac{1}{D+2q-2} \\[2mm] \dfrac{(1-\alpha^2)(\alpha^2(D+2q-2)-1)}{2\alpha^2 Dq - \alpha^4 D(D+2q-2)-1} & \text{if } \dfrac{1}{D+2q-2} < \alpha^2 < \dfrac{2q-1}{D+2q-2} \\[2mm] 1 & \text{if } \alpha^2 \geq \dfrac{2q-1}{D+2q-2} \end{cases} \quad . \tag{A.8}$$

*Proof.* After plugging Eq. (10) into Eq. (9), we have

$$L(\lambda) = \|\nabla f(x)\|_2^2 \Big( 1 - \frac{(\lambda\alpha^2 + \frac{1-\lambda}{D-1}(1-\alpha^2))^2}{(1-\frac{1}{q})(\lambda^2\alpha^2 + (\frac{1-\lambda}{D-1})^2(1-\alpha^2)) + \frac{1}{q}(\lambda\alpha^2 + \frac{1-\lambda}{D-1}(1-\alpha^2))} \Big).$$

To minimize $L(\lambda)$, we should maximize

$$F(\lambda) = \frac{(\lambda\alpha^2 + \frac{1-\lambda}{D-1}(1-\alpha^2))^2}{(1-\frac{1}{q})(\lambda^2\alpha^2 + (\frac{1-\lambda}{D-1})^2(1-\alpha^2)) + \frac{1}{q}(\lambda\alpha^2 + \frac{1-\lambda}{D-1}(1-\alpha^2))}. \tag{A.9}$$

Note that $F(\lambda)$ is a quadratic rational function w.r.t. $\lambda$.

Since we optimize $\lambda$ in a closed interval $[0, 1]$, checking $\lambda = 0$, $\lambda = 1$ and the stationary points (such that $F'(\lambda) = 0$) would suffice. By solving $F'(\lambda) = 0$, we have at most two solutions:

$$\lambda_1 = \frac{(1-\alpha^2)(\alpha^2(D+2q-2)-1)}{2\alpha^2 Dq - \alpha^4 D(D+2q-2)-1}, \tag{A.10}$$

$$\lambda_2 = \frac{1-\alpha^2}{1-\alpha^2 D},$$

where $\lambda_1$ or $\lambda_2$ is the solution if and only if the denominator is not 0. Given $\alpha^2 \leq 1$ and $D \geq 2$, $\lambda_2 \notin (0, 1)$, so we only need to consider $\lambda_1$.

First, we figure out when $\lambda_1 \in (0, 1)$. We can verify that $\lambda_1 = 1$ when $\alpha^2 = 0$ and $\lambda_1 = 0$ when $\alpha^2 = 1$. Suppose $\alpha^2 \in (0, 1)$. Let $J$ denote the numerator in Eq. (A.10) and $K$ denote the denominator. We have that when $\alpha^2 > \frac{1}{D+2q-2}$, $J > 0$; else, $J \leq 0$. We also have that when $\alpha^2 < \frac{2q-1}{D+2q-2}$, $J < K$; else, $J \geq K$. Note that $J/K \in (0, 1)$ if and only if $0 < J < K$ or $0 > J > K$. Hence, $\lambda_1 \in (0, 1)$ if and only if $\frac{1}{D+2q-2} < \alpha^2 < \frac{2q-1}{D+2q-2}$.

Case 1: $\lambda_1 \notin (0, 1)$. Then it suffices to compare $F(0)$ with $F(1)$. We have

$$F(0) = \frac{(1-\alpha^2)q}{D+q-2}, F(1) = \alpha^2.$$

Hence, $F(0) \geq F(1)$ if and only if $\alpha^2 \leq \frac{q}{D+2q-2}$. It means that if $\alpha^2 \geq \frac{2q-1}{D+2q-2}$, then $\lambda^* = 1$; if $\alpha^2 \leq \frac{1}{D+2q-2}$, then $\lambda^* = 0$.

Case 2: $\lambda_1 \in (0, 1)$. After plugging Eq. (A.10) to Eq. (A.9), we have

$$F(\lambda_1) = \frac{4\alpha^2(1-\alpha^2)(q-1)q}{-1 + 2\alpha^2(D(2q-1) + 2(q-1)^2) - \alpha^4(D+2q-2)^2}. \tag{A.11}$$

Now we prove that $F(\lambda_1) \geq F(0)$ and $F(\lambda_1) \geq F(1)$. Since when $0 < \lambda < 1$, both the numerator and the denominator in Eq. (A.9) is positive, we have $F(\lambda) > 0$, $\forall \lambda \in (0, 1)$. Since the numerator in Eq. (A.11) is non-negative and $F(\lambda_1) > 0$, we know that the denominator in Eq. (A.11) is positive. Hence, we have

$$F(\lambda_1) - F(0) = \frac{q(1-\alpha^2)(\alpha^2(D+2q-2)-1)^2}{(q+D-2)(-1+2\alpha^2(D(2q-1)+2(q-1)^2)-\alpha^4(D+2q-2)^2)} > 0;$$

$$F(\lambda_1) - F(1) = \frac{\alpha^2(\alpha^2(D+2q-2)+1-2q)^2}{-1+2\alpha^2(D(2q-1)+2(q-1)^2)-\alpha^4(D+2q-2)^2} > 0.$$

Hence in this case $\lambda^* = \lambda_1$.

The proof is complete. □

## A.4 Monotonicity of $\lambda^*$

We will prove that $\lambda^*$ is a monotonically increasing function of $\alpha^2$, and a monotonically decreasing function of $q$ (when $\alpha^2 > \frac{1}{D}$).

*Proof.* To find the monotonicity w.r.t. $\alpha^2$, note that $\lambda^* = 0$ if $\alpha^2 \leq \frac{1}{D+2q-2}$ and $\lambda^* = 1$ when $\alpha^2 \geq \frac{2q-1}{D+2q-2}$. When $\frac{1}{D+2q-2} < \alpha^2 < \frac{2q-1}{D+2q-2}$, we have

$$
\begin{aligned}
\lambda^* &= \frac{(1-\alpha^2)(\alpha^2(D+2q-2)-1)}{2\alpha^2 Dq - \alpha^4 D(D+2q-2) - 1} \\
&= \frac{\alpha^4(D+2q-2) - \alpha^2(D+2q-1) + 1}{\alpha^4 D(D+2q-2) - 2\alpha^2 Dq + 1} \\
&= \frac{1}{D}\left(1 - \frac{(\alpha^2 D - 1)(D-1)}{\alpha^4 D(D+2q-2) - 2\alpha^2 Dq + 1}\right) \\
&= \frac{1}{D} - \frac{D-1}{\alpha^2 D(D+2q-2) - (2Dq - D - 2q + 2) - 2\frac{(D-1)(q-1)}{\alpha^2 D - 1}}.
\end{aligned}
$$
(A.12)

When $\alpha^2 < \frac{1}{D}$, or when $\alpha^2 > \frac{1}{D}$, a larger $\alpha^2$ leads to larger values of both $\alpha^2 D(D+2q-2)$ and $-2\frac{(D-1)(q-1)}{\alpha^2 D - 1}$, and consequently leads to a larger $\lambda^*$. Meanwhile, by the argument in the proof of Eq. (12), when $\frac{1}{D+2q-2} < \alpha^2 < \frac{2q-1}{D+2q-2}$, the denominator of Eq. (A.10) is positive, hence $\alpha^4 D(D+2q-2) - 2\alpha^2 Dq + 1 < 0$. By Eq. (A.12), when $\alpha^2 < \frac{1}{D}$, $\lambda^* < \frac{1}{D}$; when $\alpha^2 = \frac{1}{D}$, $\lambda^* = \frac{1}{D}$; when $\alpha^2 > \frac{1}{D}$, $\lambda^* > \frac{1}{D}$. We conclude that $\lambda^*$ is a monotonically increasing function of $\alpha^2$.

To find the monotonicity w.r.t $q$ when $\alpha^2 > \frac{1}{D}$, Eq. (A.8) tells us that when $q \leq \frac{\alpha^2(D-2)+1}{2(1-\alpha^2)}$, $\lambda^* = 1$; else, $0 < \lambda^* < 1$. In the latter case, we rewrite Eq. (A.12) as

$$
\lambda^* = \frac{1}{D}\left(1 + \frac{(\alpha^2 D - 1)(D-1)}{2\alpha^2 D(1-\alpha^2)q - \alpha^4 D(D-2) - 1}\right).
$$

We have $(\alpha^2 D - 1)(D-1) > 0$, and as explained before, the denominator is positive for any $q$ such that $0 < \lambda^* < 1$. Hence, when $\alpha^2 > \frac{1}{D}$, $\lambda^*$ is a monotonically decreasing function of $q$. $\square$

## A.5 Proof of Eq. (18)

Let $\alpha = v^\top \overline{\nabla f(x)}$, $A^2 = \sum_{i=1}^d (v_i^\top \overline{\nabla f(x)})^2$. Suppose $\alpha^2 \leq 1$, $d \geq 1$, $q \geq 1$. After plugging Eq. (17) into Eq. (9), the optimal $\lambda$ is given by

$$
\lambda^* = \begin{cases} 0 & \text{if } \alpha^2 < \dfrac{A^2}{d+2q-2} \\ \dfrac{A^2(A^2 - \alpha^2(d+2q-2))}{A^4 + \alpha^4 d^2 - 2A^2\alpha^2(q+dq-1)} & \text{if } \dfrac{A^2}{d+2q-2} \leq \alpha^2 < \dfrac{A^2(2q-1)}{d} \\ 1 & \text{if } \alpha^2 \geq \dfrac{A^2(2q-1)}{d} \end{cases}.
$$

*Proof.* The proof is very similar to that in Sec. A.3. After plugging Eq. (17) into Eq. (9), we have

$$
L(\lambda) = \|\nabla f(x)\|_2^2\left(1 - \frac{(\lambda\alpha^2 + \frac{1-\lambda}{d}A^2)^2}{(1-\frac{1}{q})(\lambda^2\alpha^2 + (\frac{1-\lambda}{d})^2 A^2) + \frac{1}{q}(\lambda\alpha^2 + \frac{1-\lambda}{d}A^2)}\right).
$$

To minimize $L(\lambda)$, we should maximize

$$
F(\lambda) = \frac{(\lambda\alpha^2 + \frac{1-\lambda}{d}A^2)^2}{(1-\frac{1}{q})(\lambda^2\alpha^2 + (\frac{1-\lambda}{d})^2 A^2) + \frac{1}{q}(\lambda\alpha^2 + \frac{1-\lambda}{d}A^2)}.
$$
(A.13)

Note that $F(\lambda)$ is a quadratic rational function w.r.t. $\lambda$.

Since we optimize $\lambda$ in a closed interval $[0, 1]$, checking $\lambda = 0$, $\lambda = 1$ and the stationary points (i.e., $F'(\lambda) = 0$) would suffice. By solving $F'(\lambda) = 0$, we have at most two solutions:

$$\lambda_1 = \frac{A^2(\alpha^2(d + 2q - 2) - A^2)}{2A^2\alpha^2(dq + q - 1) - \alpha^4 d^2 - A^4}, \tag{A.14}$$

$$\lambda_2 = \frac{A^2}{A^2 - \alpha^2 d},$$

where $\lambda_1$ or $\lambda_2$ is the solution if and only if the denominator is not 0. $\lambda_2 \notin (0, 1)$, so we only need to consider $\lambda_1$.

First, we figure out when $\lambda_1 \in (0, 1)$. We can verify that $\lambda_1 = 1$ when $\alpha^2 = 0$ and $\lambda_1 = 0$ when $A^2 = 0$. Suppose $\alpha^2 \neq 0$ and $A^2 \neq 0$. Let $J$ denote the numerator in Eq. (A.14) and $K$ denote the denominator. We have that when $\alpha^2 > \frac{A^2}{d+2q-2}$, $J > 0$; else, $J \leq 0$. We also have that when $\alpha^2 < \frac{A^2(2q-1)}{d}$, $J < K$; else, $J \geq K$. Note that $J/K \in (0, 1)$ if and only if $0 < J < K$ or $0 > J > K$. Hence, $\lambda_1 \in (0, 1)$ if and only if $\frac{A^2}{d+2q-2} < \alpha^2 < \frac{A^2(2q-1)}{d}$.

Case 1: $\lambda_1 \notin (0, 1)$. Then it suffices to compare $F(0)$ and $F(1)$. We have

$$F(0) = \frac{A^2 q}{d + q - 1}, F(1) = \alpha^2.$$

Hence, $F(0) \geq F(1)$ if and only if $\alpha^2 \leq \frac{A^2 q}{d+q-1}$. It means that if $\alpha^2 \geq \frac{A^2(2q-1)}{d}$, then $\lambda^* = 1$; if $\alpha^2 \leq \frac{A^2}{d+2q-2}$, then $\lambda^* = 0$.

Case 2: $\lambda_1 \in (0, 1)$. After plugging Eq. (A.14) into Eq. (A.13), we have

$$F(\lambda_1) = \frac{4A^2\alpha^2(A^2 + \alpha^2)(q-1)q}{2A^2\alpha^2(2q(d+q-1) - d) - \alpha^4 d^2 - A^4}. \tag{A.15}$$

Now we prove that $F(\lambda_1) \geq F(0)$ and $F(\lambda_1) \geq F(1)$. Since when $0 < \lambda < 1$, both the numerator and the denominator in Eq. (A.13) is positive, we have $F(\lambda) > 0$, $\forall \lambda \in (0, 1)$. Since the numerator in Eq. (A.15) is non-negative, and $F(\lambda_1) > 0$, we know that the denominator in Eq. (A.15) is positive. Hence, we have

$$F(\lambda_1) - F(0) = \frac{qA^2(\alpha^2(d+2q-2) - A^2)^2}{(q+d-1)(2A^2\alpha^2(2q(d+q-1) - d) - \alpha^4 d^2 - A^4)} > 0;$$

$$F(\lambda_1) - F(1) = \frac{\alpha^2(\alpha^2 d + A^2(1-2q))^2}{2A^2\alpha^2(2q(d+q-1) - d) - \alpha^4 d^2 - A^4} > 0.$$

Hence in this case $\lambda^* = \lambda_1$.

The proof is complete. $\qquad\square$

### A.6   Explanation on Eq. (19)

We explain why the construction of $u_i$ in Eq. (19) makes $\mathbb{E}[u_i u_i^\top]$ a good approximation of $\mathbf{C}$.

Recall the setting: In $\mathbb{R}^D$, we have a normalized transfer gradient $v$, and a specified $d$-dimensional subspace with $\{v_1, ..., v_d\}$ as its orthonormal basis. Let $\mathbf{C} = \lambda v v^\top + \frac{1-\lambda}{d}\sum_{i=1}^d v_i v_i^\top$. Here we argue that if $u = \sqrt{\lambda} \cdot v + \sqrt{1-\lambda} \cdot \overline{(\mathbf{I} - vv^\top)\mathbf{V}\xi}$, then $\mathbb{E}[uu^\top] \approx \mathbf{C}$.

Let $r \triangleq \overline{(\mathbf{I} - vv^\top)\mathbf{V}\xi}$. The reason why $\mathbb{E}[uu^\top] \neq \mathbf{C}$ is that $\mathbb{E}[rr^\top] \neq \frac{1}{d}\sum_{i=1}^d v_i v_i^\top$ when $v$ is not orthogonal to the subspace spanned by $\{v_1, ..., v_d\}$. However, by symmetry, we still have $\mathbb{E}[r] = 0$. To get an expression of $\mathbb{E}[rr^\top]$, we let $v_T$ denotes the projection of $v$ onto the subspace, and let $v_1 = \overline{v_T}$ so that $v_2, ..., v_d$ are orthonormal to $v_T$ (hence also orthonormal to $v$). We temporarily assume $v_T \neq v$ and $v_T \neq 0$. Now let $v_1' = \overline{(\mathbf{I} - vv^\top)v_T} = \overline{v_T - v^\top v_T \cdot v}$, then $\{v_1', v_2, ..., v_d\}$ form an orthonormal basis of the subspace in which $r$ lies, and $v$ is orthogonal to this modified subspace. Now we have $\mathbb{E}[rr^\top] = \lambda_1 v_1' v_1'^\top + \frac{1-\lambda_1}{d-1}\sum_{i=2}^d v_i v_i^\top$ where $\lambda_1$ is a number in $[0, \frac{1}{d}]$. (Note

that when $v = v_T$, although $v'_1$ cannot be defined, we have $\lambda_1 = 0$. When $v_T = 0$, we can just set $v'_1 = v_1$ and $\lambda_1 = \frac{1}{d}$.) When $d$ is large, $\lambda_1$ is small, so for approximation we can replace $v'_1$ with $v_1$; $|\lambda - \frac{1}{d}|$ is small, so for approximation we can set $\lambda_1 = \frac{1}{d}$. Then we have $\mathbb{E}[rr^\top] \approx \frac{1}{d} \sum_{i=1}^d v_i v_i^\top$. Since $\mathbb{E}[r] = 0$, we have $\mathbb{E}[uu^\top] = \lambda vv^\top + (1-\lambda)\mathbb{E}[rr^\top] \approx \lambda vv^\top + \frac{1-\lambda}{d} \sum_{i=1}^d v_i v_i^\top$.

**Remark 3.** *To avoid approximation, one can choose the subspace as spanned by $\{v'_1, v_2, ..., v_d\}$ instead of $\{v_1, v_2, ..., v_d\}$ to ensure that $v$ is orthogonal to the subspace. Then $u$ can be sampled as*

$$u = \sqrt{\lambda} \cdot v + \sqrt{1-\lambda} \cdot \overline{\mathbf{V}'\xi},$$

*where $\mathbf{V}' = [v'_1, v_2, ..., v_d]$ and $\xi$ is sampled uniformly from the $d$-dimensional unit hypersphere. Note that here the optimal $\lambda$ is calculated using $A'^2 = v'^\top_1 \overline{\nabla f(x)} + \sum_{i=2}^d (v_i^\top \overline{\nabla f(x)})^2$. However, in practice, it is not convenient to make the subspace dependent on $v$, and the computational complexity is high to construct an orthonormal basis with one vector ($v'_1$) specified.*

## B   Gradient averaging method

In Sec. 3.2, we have presented the prior-guided random gradient-free (P-RGF) algorithm, where we integrate the transfer gradient into the sampling distribution of $u_i$. In this section, we propose the gradient averaging algorithm as an alternative method to incorporate the transfer gradient. The motivation is as follows. We observe that the RGF estimator in Eq. (5) is in the following form: $\hat{g} = \frac{1}{q} \sum_{i=1}^q \hat{g}_i$, where multiple rough estimates are averaged. Indeed, the transfer gradient itself can also be considered as an estimate of the true gradient, and then it is reasonable to adopt a weighted average of the transfer gradient and the RGF estimator. Here, we choose the RGF estimator to be the ordinary one (using $u_i$ sampled from uniform distribution) instead of the P-RGF estimator, to prevent its direction from being too similar to the direction of the transfer gradient.

In summary, the gradient averaging method works as follows. We first get the RGF estimator denoted by $\hat{g}^U$, given by Eq. (5) with the sampling distribution $\mathcal{P}$ being the uniform distribution; then normalize the estimator; and finally average the normalized transfer gradient $v$ and the normalized RGF estimator $\overline{\hat{g}^U}$ as

$$\hat{g} = (1-\mu)v + \mu\overline{\hat{g}^U}, \tag{B.1}$$

where $\mu \in [0, 1]$ plays a similar role as $\lambda$ in the proposed prior-guided RGF method. We also assume $\alpha = v^\top \nabla f(x) \geq 0$. Under the gradient estimation problem, we also want to minimize $L(\hat{g})$ by optimizing $\mu$. First, we have the following theorem.

**Theorem 2.** *Let $\beta = \overline{\nabla f(x)}^\top \frac{1}{q} \sum_{i=1}^q (u_i^\top \nabla f(x) \cdot u_i)$ be the cosine similarity between $\nabla f(x)$ and the ordinary RGF estimator w.r.t. a locally linear $f$. If $f$ is differentiable at $x$, the loss of the gradient estimator in Eq. (B.1) is*

$$\lim_{\sigma \to 0} L(\hat{g}) = (1 - \frac{(\mu\alpha + (1-\mu)\mathbb{E}[\beta])^2}{\mu^2 + (1-\mu)^2 + 2\mu(1-\mu)\alpha\mathbb{E}[\beta]})\|\nabla f(x)\|_2^2. \tag{B.2}$$

*Proof.* As in Eq. (5), $\hat{g}^U = \frac{1}{q} \sum_{i=1}^q \hat{g}_i^U$ and $\hat{g}_i^U = \frac{f(x+\sigma u_i) - f(x)}{\sigma} \cdot u_i$. First, we derive $L(\hat{g})$ based on the assumption that $\hat{g}_i^U$ is equal to $u_i^\top \nabla f(x) \cdot u_i$, which will hold when $f$ is locally linear.

**Lemma 4.** *Assume that $\hat{g}^U = \frac{1}{q} \sum_{i=1}^q (u_i^\top \nabla f(x) \cdot u_i)$ (then $\beta = \overline{\nabla f(x)}^\top \overline{\hat{g}^U}$). We have*

$$L(\hat{g}) = (1 - \frac{(\mu\alpha + (1-\mu)\mathbb{E}[\beta])^2}{\mu^2 + (1-\mu)^2 + 2\mu(1-\mu)\alpha\mathbb{E}[\beta]})\|\nabla f(x)\|_2^2.$$

*Proof.* It can be verified[1] that $\hat{g}^U = 0$ happens with probability 0, hence we restrict our consideration to the set $\{\hat{g}^U \neq 0\}$, which does not affect our conclusion. Then $\overline{\hat{g}^U}$ is always well-defined. The distribution of $\hat{g}^U$ is symmetric around the direction of $\nabla f(x)$, and so is the distribution of

$\overline{\hat{g}^U}$. Hence we can suppose that $\mathbb{E}[\overline{\hat{g}^U}] = k\overline{\nabla f(x)}$. Since $\mathbb{E}[\beta] = \overline{\nabla f(x)}^\top \mathbb{E}[\overline{\hat{g}^U}] = k$, we have $\mathbb{E}[\overline{\hat{g}^U}] = \mathbb{E}[\beta]\overline{\nabla f(x)}$.

Hence we have

$$\nabla f(x)^\top \mathbb{E}[\overline{\hat{g}^U}] = \nabla f(x)^\top \mathbb{E}[\beta]\overline{\nabla f(x)} = \mathbb{E}[\beta]\|\nabla f(x)\|_2,$$

and

$$v^\top \mathbb{E}[\overline{\hat{g}^U}] = v^\top \mathbb{E}[\beta]\overline{\nabla f(x)} = \alpha\mathbb{E}[\beta].$$

Together with $v^\top \nabla f(x) = \alpha\|\nabla f(x)\|_2$ and noting that $\|v\|_2 = 1$, we have

$$\begin{aligned}
\mathbb{E}\|\nabla f(x) - b\hat{g}\|_2^2 &= \mathbb{E}\|b\mu v + b(1-\mu)\overline{\hat{g}^U} - \nabla f(x)\|^2 \\
&= b^2\mu^2 + b^2(1-\mu)^2 + \|\nabla f(x)\|_2^2 + 2b^2\mu(1-\mu)v^\top\mathbb{E}[\overline{\hat{g}^U}] \\
&\quad - 2b\mu\alpha\|\nabla f(x)\|_2 - 2b(1-\mu)\nabla f(x)^\top\mathbb{E}[\overline{\hat{g}^U}] \qquad \text{(B.3)} \\
&= b^2\mu^2 + b^2(1-\mu)^2 + \|\nabla f(x)\|_2^2 + 2b^2\mu(1-\mu)\alpha\mathbb{E}[\beta] \\
&\quad - 2b\mu\alpha\|\nabla f(x)\|_2 - 2b(1-\mu)\mathbb{E}[\beta]\|\nabla f(x)\| \\
&= ((1-\mu)^2 + \mu^2 + 2\mu(1-\mu)\alpha\mathbb{E}[\beta])b^2 \\
&\quad - 2(\alpha\mu + \mathbb{E}[\beta](1-\mu))\|\nabla f(x)\|_2 b + \|\nabla f(x)\|_2^2.
\end{aligned}$$

Since $\nabla f(x)^\top \hat{g}^U = \frac{1}{q}\sum_{i=1}^q (u_i^\top \nabla f(x))^2 \geq 0$, then $\beta \geq 0$, and hence $\mathbb{E}[\beta] \geq 0$. Then $(1-\mu)^2 + \mu^2 + 2\mu(1-\mu)\alpha\mathbb{E}[\beta] > 0$ and $\alpha\mu + \mathbb{E}[\beta](1-\mu) \geq 0$. Since $L(\hat{g}) = \min_{b \geq 0} \mathbb{E}\|\nabla f(x) - b\hat{g}\|_2^2$, optimize the objective w.r.t. $b$ and we complete the proof. $\qquad\square$

Next, we prove that if $f$ is not locally linear, as long as it is differentiable at $x$, then by picking a sufficient small $\sigma$, the loss tends to be that of the local linear approximation. Here, we redefine the notation as follows. We make the dependency of $\hat{g}^U$ on $\sigma$ explicit, i.e. we use $\hat{g}_\sigma^U$ to denote it. Meanwhile, we define $\hat{g}_0^U \triangleq \frac{1}{q}\sum_{i=1}^q (u_i^\top \nabla f(x) \cdot u_i)$ as the RGF estimator under the local linear approximation. We define $\hat{g}_\sigma = (1-\mu)v + \mu\hat{g}_\sigma^U$ and $\hat{g}_0 = (1-\mu)v + \mu\hat{g}_0^U$. Then we have

**Lemma 5.** *If $f$ is differentiable at $x$, then*

$$\lim_{\sigma \to 0} L(\hat{g}_\sigma) = L(\hat{g}_0)$$

*Proof.* By Eq. (B.3), it suffices to prove $\lim_{\sigma \to 0} \mathbb{E}[\overline{\hat{g}_\sigma^U}] = \mathbb{E}[\overline{\hat{g}_0^U}]$.

For any value of $u_1, u_2, ..., u_q$, we have $\lim_{\sigma \to 0} \hat{g}_\sigma^U = \hat{g}_0^U$, i.e. $\hat{g}_\sigma^U$ converges pointwise to $\hat{g}_0^U$. Recall that $\Pr(\hat{g}_0^U = 0) = 0$, so we can restrict our consideration to the set $\{\hat{g}_0^U \neq 0\}$ which does not affect our conclusion. Since $\overline{x} = \frac{x}{\|x\|_2}$ is continuous everywhere in its domain, $\overline{\hat{g}_\sigma^U}$ converges pointwise to $\overline{\hat{g}_0^U}$. Since the family $\{\overline{\hat{g}_\sigma^U}\}$ is uniformly bounded, by dominated convergence theorem we have $\lim_{\sigma \to 0} \mathbb{E}[\overline{\hat{g}_\sigma^U}] = \mathbb{E}[\overline{\hat{g}_0^U}]$. $\qquad\square$

By combining the two lemmas above, our proof for the theorem is complete. $\qquad\square$

We can calculate the closed-form solution of $\mu^*$, the value of $\mu$ minimizing Eq. (B.2), as

$$\mu^* = \frac{(1-\alpha^2)\mathbb{E}[\beta]}{(1-\alpha^2)\mathbb{E}[\beta] + \alpha(1-\mathbb{E}[\beta]^2)} \approx \frac{\mathbb{E}[\beta]}{\mathbb{E}[\beta] + \alpha}. \qquad \text{(B.4)}$$

That is, the ratio of weights of $v$ and $\overline{\hat{g}^U}$ is approximately the ratio of their (expected) inner product with the true gradient.

Next, we discuss how to calculate $\mathbb{E}[\beta] = \mathbb{E}[\overline{\nabla f(x)}^\top \overline{\hat{g}_0^U}]$, where $\hat{g}_0^U = \frac{1}{q}\sum_{i=1}^q (u_i^\top \nabla f(x) \cdot u_i)$. $\mathbb{E}[\beta]$ is independent of $\|\nabla f(x)\|_2$, and since $u_i$ is uniformly sampled from the unit hypersphere, $\mathbb{E}[\beta]$ is also independent of the direction of $\nabla f(x)$. Hence, $\mathbb{E}[\beta]$ is a constant given the dimension $D$ and the number of queries $q$, and we can estimate $\mathbb{E}[\beta]$ using numerical simulation methods.

---

**Algorithm 1** Gradient averaging method

---

**Input:** The black-box model $f$; input $x$ and label $y$; the normalized transfer gradient $v$; sampling variance $\sigma$; number of queries $q$; input dimension $D$; threshold $c$.
**Output:** Estimate of the gradient $\nabla f(x)$.
 1: Estimate the cosine similarity $\alpha = v^\top \overline{\nabla f(x)}$ (detailed in Sec. 3.3);
 2: Approximate $\mathbb{E}[\beta]$ as $\sqrt{\frac{q}{D+q-1}}$;
 3: Calculate $\mu^*$ according to Eq. (B.4) given $\alpha$ and $\mathbb{E}[\beta]$;
 4: **if** $\mu^* \leq c$ **then**
 5: $\quad$ **return** $v$;
 6: **end if**
 7: $\hat{g}^U \leftarrow \mathbf{0}$;
 8: **for** $i = 1$ to $q$ **do**
 9: $\quad$ Sample $u_i$ from the uniform distribution on the $D$-dimensional unit hypersphere;
 10: $\quad \hat{g}^U \leftarrow \hat{g}^U + \dfrac{f(x + \sigma u_i, y) - f(x, y)}{\sigma} \cdot u_i$;
 11: **end for**
 12: **return** $\nabla f(x) \leftarrow (1 - \mu^*)v + \mu^* \overline{\hat{g}^U}$.

---

However, here we give a framework for approximating $\mathbb{E}[\beta]$ in a closed-form formula. We notice that the following approximation works well in practice, where $\hat{g} = \frac{1}{q} \sum_{i=1}^{q} (u_i^\top \nabla f(x) \cdot u_i)$:

$$
\begin{aligned}
\mathbb{E}[\beta] &= \mathbb{E}[\sqrt{\beta^2}] \\
&\approx \sqrt{\mathbb{E}[\beta^2]} \\
&= \sqrt{1 - \mathbb{E}[\min_b \|\overline{\nabla f(x)} - b\hat{g}\|^2]} \\
&= \sqrt{1 - \frac{1}{\|\nabla f(x)\|_2^2} \mathbb{E}[\min_b \|\overline{\nabla f(x)} - b\hat{g}\|^2]} \\
&\approx \sqrt{1 - \frac{1}{\|\nabla f(x)\|_2^2} \min_b \mathbb{E}\|\overline{\nabla f(x)} - b\hat{g}\|^2} \\
&= \sqrt{1 - \frac{1}{\|\nabla f(x)\|_2^2} L(\hat{g})^2}.
\end{aligned}
$$

Here, the first equality is because $\nabla f(x)^\top \hat{g} = \frac{1}{q} \sum_{i=1}^{q} (u_i^\top \nabla f(x))^2 \geq 0$; the second equality is because we have $\min_b \|\overline{\nabla f(x)} - b\hat{g}\|^2 = 1 - (\overline{\nabla f(x)}^\top \overline{\hat{g}})^2 = 1 - \beta^2$. Intuitively, the two approximations work well because the variances of $\beta$ and $\|\hat{g}\|_2$ are relatively small.

Now we define $F(\hat{g}) = 1 - \frac{1}{\|\nabla f(x)\|_2^2} L(\hat{g})^2$. Then we have $\mathbb{E}[\beta] \approx \sqrt{F(\hat{g})}$. Note that when $u_i$ is sampled from the uniform distribution on the unit hypersphere, $F(\hat{g})$ is in fact $F(\frac{1}{D})$ in Eq. (A.9), since $\hat{g}$ is an RGF estimator w.r.t. locally linear $f$, and $\mathbb{E}[u_i u_i^\top] = \frac{1}{D}\mathbf{I}$ which corresponds to $\lambda = \frac{1}{D}$ in Eq. (10). We can calculate $F(\frac{1}{D}) = \frac{q}{D+q-1}$. Hence, $\mathbb{E}[\beta] \approx \sqrt{\frac{q}{D+q-1}}$.

Calculating $\mu^*$ using $\alpha \geq 0$ and $\mathbb{E}[\beta] \approx \sqrt{\frac{q}{D+q-1}} > 0$, we have $\mu^* > 0$. This means we always need to take $q$ queries to get $\overline{\hat{g}^U}$. However, when $\mu$ is close to 0, the improvement of using $\hat{g} = (1 - \mu^*)v + \mu^* \overline{\hat{g}^U}$ instead of directly using $v$ as the estimate is marginal. To save queries, we adopt a threshold $c \in (0, 1)$. When $\mu^* \leq c$, we let $\hat{g} = v$ instead of letting $\hat{g} = (1 - \mu^*)v + \mu^* \overline{\hat{g}^U}$.

We summarize the gradient averaging method in Algorithm 1.

## B.1 Incorporating the data-dependent prior

We can also incorporate the data-dependent prior introduced in Sec. 3.4 into the proposed gradient averaging method. In this case, we get an ordinary subspace RGF estimate $\hat{g}^S$ first[2] (instead of an ordinary RGF estimate); and then normalize it; and finally get the averaged estimator as

$$\hat{g} = (1 - \mu)v + \mu \overline{\hat{g}^S}. \tag{B.5}$$

We also assume $\alpha = v^\top \overline{\nabla f(x)} \geq 0$. Here, we need to analyze some quantity about the subspace. We define $\overline{\nabla f(x)}_T = (\sum_{i=1}^d v_i v_i^\top) \overline{\nabla f(x)}$ is the projection of $\overline{\nabla f(x)}$ into the subspace corresponding to the data-dependent prior, and $A^2 = \sum_{i=1}^d (v_i^\top \overline{\nabla f(x)})^2 = \|\overline{\nabla f(x)}_T\|^2$. Then we have the following loss function:

**Theorem 3.** *Let $\beta = \overline{\nabla f(x)}^\top \frac{1}{q} \sum_{i=1}^q (u_i^\top \nabla f(x) \cdot u_i)$ be the cosine similarity between $\nabla f(x)$ and the ordinary subspace RGF estimator w.r.t. a locally linear $f$. (Note that here $u_i$ lies in the subspace.) Further more, let $\alpha_1 = v^\top \overline{\nabla f(x)}_T$. If $f$ is differentiable at $x$ and $A^2 > 0$, using $\hat{g}$ defined in (B.5), we have*

$$L(\hat{g}) = (1 - \frac{(\mu\alpha + (1-\mu)\mathbb{E}[\beta])^2}{\mu^2 + (1-\mu)^2 + 2\mu(1-\mu)\frac{\alpha_1}{A^2}\mathbb{E}[\beta]})\|\nabla f(x)\|^2. \tag{B.6}$$

*Proof.* Similar to the proof of Theorem 2, we define $\hat{g}_0^S = \frac{1}{q}\sum_{i=1}^q (u_i^\top \nabla f(x) \cdot u_i) = \frac{1}{q}\sum_{i=1}^q (u_i^\top \nabla f(x)_T \cdot u_i)$, where $\nabla f(x)_T = \|\nabla f(x)\|_2 \overline{\nabla f(x)}_T$ denotes the projection of $\nabla f(x)$ into the subspace. Then $\beta = \overline{\nabla f(x)}^\top \hat{g}_0^S = \overline{\nabla f(x)}_T^\top \hat{g}_0^S$. Since $A^2 > 0$, we have $\nabla f(x)_T \neq 0$, hence as described in Footnote 1, we can prove $\Pr(\hat{g}_0^S = 0) = 0$ similarly. Now we restrict our consideration to the set $\{\hat{g}_0^S \neq 0\}$. The distribution of $\hat{g}_0^S$ is symmetric around the direction of $\nabla f(x)_T$, and so is the distribution of $\overline{\hat{g}_0^S}$. Hence we can suppose that $\mathbb{E}[\overline{\hat{g}_0^S}] = k\overline{\nabla f(x)}_T$. Since $\mathbb{E}[\beta] = \overline{\nabla f(x)}_T^\top \mathbb{E}[\overline{\hat{g}_0^S}] = k\|\overline{\nabla f(x)}_T\|_2^2 = kA^2$, we have $\mathbb{E}[\overline{\hat{g}_0^S}] = \frac{\mathbb{E}[\beta]}{A^2}\overline{\nabla f(x)}_T$.

Note that

$$v^\top \mathbb{E}[\overline{\hat{g}_0^S}] = v^\top \frac{\mathbb{E}[\beta]}{A^2}\overline{\nabla f(x)}_T = \frac{\alpha_1}{A^2}\mathbb{E}[\beta].$$

The rest of the proof is the same as that of Theorem 2. □

The optimal solution of $\mu$ minimizing Eq. (B.6) is

$$\mu^* = \frac{(A^2 - \alpha_1\alpha)\mathbb{E}[\beta]}{(A^2 - \alpha_1\mathbb{E}[\beta])(\alpha + \mathbb{E}[\beta])} \approx \frac{\mathbb{E}[\beta]}{\mathbb{E}[\beta] + \alpha}. \tag{B.7}$$

Hence, the approximate solution is the same as in the case without using the data-dependent prior, which does not depend on $\alpha_1$.

Similarly, we can approximate $\mathbb{E}[\beta]$ by $\mathbb{E}[\beta] \approx \sqrt{F(\hat{g})}$. When $u_i$ is sampled from the uniform distribution on the unit hypersphere in the subspace, $F(\hat{g})$ is in fact $F(0)$ in Eq. (A.13), since $\hat{g}$ is an RGF estimator w.r.t. locally linear $f$, and $\mathbb{E}[u_i u_i^T] = \frac{1}{d}\sum_{i=1}^d v_i v_i^\top$ which corresponds to $\lambda = 0$ in Eq. (17). We can calculate $F(0) = \frac{A^2 q}{d+q-1}$. Hence, $\mathbb{E}[\beta] \approx \sqrt{\frac{A^2 q}{d+q-1}}$.

Our gradient averaging algorithm with the data-dependent prior is similar to Algorithm 1. We first estimate $\alpha$ and $A$, approximate $\mathbb{E}[\beta]$ as $\sqrt{\frac{A^2 q}{d+q-1}}$, and then calculate $\mu^*$ by Eq. (B.7). If $\mu^* \leq c$, we use the transfer gradient $v$ as the estimate. If not, we get the ordinary subspace RGF estimator $\hat{g}^S$, then use $\hat{g} \leftarrow (1 - \mu^*)v + \mu^* \overline{\hat{g}^S}$ as the estimate.

## C   Estimation of $A$

Suppose that the subspace is spanned by a set of orthonormal vectors $\{v_1, ..., v_d\}$. Now we want to estimate

$$A^2 = \sum_{i=1}^{d}(v_i^\top \overline{\nabla f(x)})^2 = \frac{\sum_{i=1}^{d}(v_i^\top \nabla f(x))^2}{\|\nabla f(x)\|_2^2} = \frac{\|h(x)\|_2^2}{\|\nabla f(x)\|_2^2},$$

where $h(x) = \sum_{i=1}^{d} v_i^\top \nabla f(x) \cdot v_i$ is the projection of $\nabla f(x)$ to the subspace. We can estimate $\|\nabla f(x)\|_2^2$ using the method introduced in Sec. 3.3. Here, we introduce the method to estimate $\|h(x)\|_2^2$.

Let $w = \mathbf{V}\xi$ where $\mathbf{V} = [v_1, v_2, ..., v_d]$ and $\xi$ is a random vector uniformly sampled from the $d$-dimensional unit hypersphere. By Lemma 3, $\mathbb{E}[ww^\top] = \frac{1}{d}\sum_{i=1}^{d} v_i v_i^\top$. Suppose we have $S$ i.i.d. such samples of $w$ denoted by $w_1, ..., w_S$, and we let $\mathbf{W} = [w_1, ..., w_S]$.

With $g(x_1, ..., x_S) = \frac{1}{S}\sum_{s=1}^{S} x_s^2$, we have

$$g(\mathbf{W}^\top \nabla f(x)) = g(\mathbf{W}^\top h(x)) = \|h(x)\|_2^2 \cdot g(\mathbf{W}^\top \overline{h(x)}).$$

Hence $\frac{g(\mathbf{W}^\top \nabla f(x))}{\mathbb{E}[g(\mathbf{W}^\top \overline{h(x)})]}$ is an unbiased estimator of $\|h(x)\|_2^2$. Now, $\overline{h(x)}$ is in the subspace spanned by $\{v_1, ..., v_d\}$, and $w_1$ is uniformly distributed on the unit hypersphere of this subspace. Hence $\mathbb{E}[(w_1^\top \overline{h(x)})^2]$ is independent of the direction of $\overline{h(x)}$ and can be computed. We have:

$$\mathbb{E}[g(\mathbf{W}^\top \overline{h(x)})] = \mathbb{E}[(w_1^\top \overline{h(x)})^2] = \overline{h(x)}^\top \mathbb{E}[w_1 w_1^\top]\overline{h(x)} = \overline{h(x)}^\top \frac{1}{d}\sum_{i=1}^{d} v_i v_i^\top \overline{h(x)} = \frac{1}{d}.$$

Hence, we have the estimator $\|h(x)\|_2 \approx \sqrt{\frac{d}{S}\sum_{s=1}^{S}(w_s^\top \nabla f(x))^2}$, where $w_s = \mathbf{V}\xi_s$ and $\xi_s$ is uniformly sampled from the unit hypersphere in $\mathbb{R}^d$. Finally we can get an estimate of $A$ by $A = \frac{\|h(x)\|_2}{\|\nabla f(x)\|_2}$.

## D   Additional experiments

We add the experimental results using the gradient averaging method, including a baseline method which uses a fixed $\mu$ set to 0.5 and the algorithm using the optimal value $\mu^*$ given by Eq. (B.4) (or by Eq. (B.7) in the case with the data-dependent prior). We set $c = \frac{1}{1+\sqrt{2}}$, and the other hyperparameters are the same with those for the P-RGF method. Table 3 and Table 4 are the full tables of experimental results based on the $\ell_2$ norm.

We show the experimental results based on the $\ell_\infty$ norm in this section. We set the perturbation budget as $\epsilon = 0.05$, the step size as $\eta = 0.005$ in the PGD method. Other hyperparameters are the same with those for $\ell_2$ attacks. Table 5 and Table 6 show the results for attacking the normal models and the defensive models, respectively. Our method also leads to better results, which are consistent with those based on the $\ell_2$ norm.

Table 3: The full experimental results of black-box attacks against Inception-v3, VGG-16, and ResNet-50 under the $\ell_2$ norm. We report the attack success rate (ASR) and the average number of queries (AVG. Q) needed to generate an adversarial example over successful attacks.

| Methods | Inception-v3 | | VGG-16 | | ResNet-50 | |
|---|---|---|---|---|---|---|
| | ASR | AVG. Q | ASR | AVG. Q | ASR | AVG. Q |
| NES [2] | 95.5% | 1718 | 98.7% | 1081 | 98.4% | 969 |
| Bandits$_T$ [3] | 92.4% | 1560 | 94.0% | 584 | 96.2% | 1076 |
| Bandits$_{TD}$ [3] | 97.2% | 874 | 94.9% | 278 | 96.8% | 512 |
| AutoZoom [5] | 85.4% | 2443 | 96.2% | 1589 | 94.8% | 2065 |
| RGF | 97.7% | 1309 | 99.8% | 935 | 99.5% | 809 |
| P-RGF ($\lambda = 0.5$) | 96.5% | 1119 | 97.3% | 1075 | 98.3% | 990 |
| P-RGF ($\lambda^*$) | **98.1%** | 745 | **99.8%** | 521 | **99.6%** | 452 |
| Averaging ($\mu = 0.5$) | 96.9% | 1140 | 94.6% | 2143 | 96.3% | 2257 |
| Averaging ($\mu^*$) | 97.9% | **735** | **99.8%** | **516** | 99.5% | **446** |
| RGF$_D$ | 99.1% | 910 | **100.0%** | 464 | **99.8%** | 521 |
| P-RGF$_D$ ($\lambda = 0.5$) | 98.2% | 1047 | 99.3% | 917 | 99.3% | 893 |
| P-RGF$_D$ ($\lambda^*$) | 99.1% | 649 | 99.7% | 370 | 99.6% | **352** |
| Averaging$_D$ ($\mu = 0.5$) | **99.2%** | 768 | 99.9% | 900 | 99.2% | 1177 |
| Averaging$_D$ ($\mu^*$) | **99.2%** | **644** | 99.8% | **366** | 99.5% | 355 |

Table 4: The full experimental results of black-box attacks against JPEG compression [1], randomization [7], and guided denoiser [4] under the $\ell_2$ norm. We report the attack success rate (ASR) and the average number of queries (AVG. Q) needed to generate an adversarial example over successful attacks.

| Methods | JPEG Compression [1] | | Randomization [7] | | Guided Denoiser [4] | |
|---|---|---|---|---|---|---|
| | ASR | AVG. Q | ASR | AVG. Q | ASR | AVG. Q |
| NES [2] | 47.3% | 3114 | 23.2% | 3632 | 48.0% | 3633 |
| SPSA [6] | 40.0% | 2744 | 9.6% | 3256 | 46.0% | 3526 |
| RGF | 41.5% | 3126 | 19.5% | 3259 | 50.3% | 3569 |
| P-RGF | 61.4% | 2419 | 60.4% | 2153 | 51.4% | 2858 |
| Averaging | **69.4%** | **2134** | **72.8%** | **1739** | **66.6%** | **2441** |
| RGF$_D$ | 70.4% | 2828 | 54.9% | 2819 | 83.7% | 2230 |
| P-RGF$_D$ | **81.1%** | 2120 | **82.3%** | 1816 | **89.6%** | 1784 |
| Averaging$_D$ | 80.6% | **2087** | 77.4% | **1700** | 87.2% | **1777** |

Table 5: The experimental results of black-box attacks against Inception-v3, VGG-16, and ResNet-50 under the $\ell_\infty$ norm. We report the attack success rate (ASR) and the average number of queries (AVG. Q) needed to generate an adversarial example over successful attacks.

| Methods | Inception-v3 | | VGG-16 | | ResNet-50 | |
|---|---|---|---|---|---|---|
| | ASR | AVG. Q | ASR | AVG. Q | ASR | AVG. Q |
| NES [2] | 87.5% | 1850 | 95.6% | 1477 | 94.5% | 1405 |
| Bandits$_T$ [3] | 89.5% | 1891 | 93.8% | 585 | 95.2% | 1199 |
| Bandits$_{TD}$ [3] | 94.7% | 1099 | 95.1% | 288 | 96.5% | 651 |
| RGF | 94.4% | 1565 | **98.8%** | 1064 | **99.4%** | 990 |
| P-RGF ($\lambda = 0.5$) | 85.4% | 1578 | 90.5% | 1420 | 92.1% | 1250 |
| P-RGF ($\lambda^*$) | 93.8% | 979 | 98.4% | 731 | 99.2% | 650 |
| Averaging ($\mu = 0.5$) | 91.8% | 1350 | 87.4% | 2453 | 88.8% | 2547 |
| Averaging ($\mu^*$) | **94.8%** | **974** | 98.4% | **685** | 99.1% | **632** |
| RGF$_D$ | 97.2% | 1034 | **100.0%** | 502 | **99.7%** | 595 |
| P-RGF$_D$ ($\lambda = 0.5$) | 91.2% | 1403 | 96.2% | 1075 | 96.4% | 1156 |
| P-RGF$_D$ ($\lambda^*$) | 97.3% | 812 | 99.6% | 433 | 99.6% | 452 |
| Averaging$_D$ ($\mu = 0.5$) | 97.6% | 948 | 99.2% | 983 | 98.3% | 1316 |
| Averaging$_D$ ($\mu^*$) | **98.4%** | **772** | 99.7% | **420** | 99.6% | **439** |

Table 6: The experimental results of black-box attacks against JPEG compression [1], randomization [7], and guided denoiser [4] under the $\ell_\infty$ norm. We report the attack success rate (ASR) and the average number of queries (AVG. Q) needed to generate an adversarial example over successful attacks.

| Methods | JPEG Compression [1] | | Randomization [7] | | Guided Denoiser [4] | |
|---|---|---|---|---|---|---|
| | ASR | AVG. Q | ASR | AVG. Q | ASR | AVG. Q |
| NES [2] | 29.9% | 2694 | 14.8% | 3027 | 20.0% | 3423 |
| SPSA [6] | 37.1% | 2775 | 10.7% | 2809 | 26.9% | 3343 |
| RGF | 27.1% | 2716 | 12.6% | 3005 | 26.0% | 3120 |
| P-RGF | 44.8% | 2491 | 41.7% | 2132 | 32.9% | 2507 |
| Averaging | **51.8%** | **2138** | **51.9%** | **1813** | **38.7%** | **2251** |
| $RGF_D$ | 53.4% | 2708 | 42.4% | 2444 | 73.3% | 2158 |
| $P\text{-}RGF_D$ | **64.0%** | 2189 | **66.9%** | 2108 | 76.0% | **1799** |
| $Averaging_D$ | **64.0%** | **2141** | 58.3% | **1753** | **77.6%** | 1889 |

## Footnotes

[1]If $\hat{g}^U = 0$, $\nabla f(x)^\top \hat{g}^U = \frac{1}{q} \sum_{i=1}^q (u_i^\top \nabla f(x))^2 = 0$, hence $u_i^\top \nabla f(x) = 0$ for $i = 1, 2, ..., q$, whose probability is 0.

[2] An ordinary subspace RGF estimate refers to the RGF estimate in Eq. (5) with $u_i = \mathbf{V}\xi_i$, where $\xi_i$ is sampled uniformly from the $d$-dimensional unit hypersphere, $\mathbf{V} = [v_1, v_2, ..., v_d]$, and $\{v_1, v_2, ..., v_d\}$ is an orthonormal basis of a $d$-dimensional subspace. It corresponds to $\lambda = 0$ in Eq. (17).