[Reviews · NeurIPS 2019]

Reviewer 1



Originality: Although there is a previous work [17] talking about the same setting, the paper have a more theoretical and general analysis, which is very good in the originality. Quality: The overall paper quality is good. However, I have a comment on the experiments on performance of gradient estimation. The average cosine similarity is so low and it seems not very useful. I would suggest to make another experiments to show the effectiveness. See the improvements. Clarity: The paper is well-written and easy to follow. Significance: It could be very useful combine the transfer-based attack with blackbox attack in the practice. Although the performance is not significantly better than the previous STOA bandit attack, it would be a good supplement. However, I have some concerns regarding the experiment results. The RGF method should be very similar or exactly same with NES. However, the results show the RGF methods always outperforms NES, which doesn’t make sense.

Reviewer 2



### Post response comments I would like to thank the authors for addressing my concerns carefully, especially on validating advantages of setting lambda adaptively over a fixed value. A regret is about estimation for cosine similarity (my concern 2). Although the response adds the specific value of S, it is still not explained **when** and **how often** to estimate cosine similarity (see line 197–198). It should have an important impact on query complexity but ignored in experiments. It is suggested to make similarity estimation clear in a final version. *** Transfer-based attack and query-based attack are two common types of black-box adversarial attack. The idea is OK to combine transfer-based attack and query-based attack. The paper proposes a simple method where the gradient of the surrogate model is used as a prior of the true gradient. My concerns are as follows. **Concern 1** My main concern lies on the novelty of the idea. It is clearly discussed in the previous paper *Guessing smart: Biased sampling for efficient black-box adversarial attacks* that the gradient of the surrogate model. The formats are almost the same except the paper proposes an "adaptive" way to set the weight $\lambda^*$. **Concern 2** To set the weight $\lambda^*$, the proposed method has to solve another esitmation problem for cosine similarity between the surrogate gradient and the true gradient (or gradient norm). The esitmiation is quite hearistic and lacks necessary analysis. It is nececessary to analyze how the estimation influences the resulted true gradient estimate. More importantly, the estimation between the surrogate gradient and the true gradient is completely neglected in the experiement section. It is necessary to empirically investigate how to estimate the gradient norm, e.g., how to set $S$. **Concern 3** When investigating whether setting $\lambda^*$ adaptively is necessary, $\lambda=0.5$ is not a good choice. As Figure 2(b) shows (it would be better to plot the distribution of $\lambda^*$.), to get a higher cosine similarity, $\lambda$ should be much smaller than $0.5$. It would be better to compare P-RGF ($\lambda=0.05 (?)$) rather than P-RGF ($\lambda=0.5 (?)$).

Reviewer 3



[Edit after the author feedback]: I thank the authors for addressing my comments during the author feedback. I have read the authors' response as well as the other reviews. The authors' response, especially the updated attack results on \ell_{\infty} adv trained models (Table B), addresses my concerns on the effectiveness of P-RGF. Overall, I think this submission is interesting and provides an efficient and effective adversarial attack approach. I am happy to raise my rating. ========================================================== Summary: To improve attack success rates and query efficiency for black-box adversarial attacks, this paper proposed a prior-guided method which is able to better estimate the gradient in a high-dimensional under the black-box scenario. Theoretically, this paper establishes the optimal coefficient of the proposed algorithm. Empirically, the proposed algorithm can use fewer queries to achieve higher attack success rates compared with previous approaches. Pros: - This paper derives the optimal coefficient, i.e., $\lambda^{*}$, in the proposed algorithmic framework, which makes the algorithm more efficient and effective. Also, the algorithm is simple and easy to implement. - The proposed attack framework is general and is able to incorporate various data-dependent prior Information. - The empirical results demonstrate the effectiveness of the proposed method, especially when attacking defensive models. Limitation & Questions: - As described in Table 1, it seems that the improvement over the RGF method is not significant. - As adversarial training, like [22], has been shown to be an effective approach to defend against adversarial attacks, it would be better to add the attack results on adversarial trained defensive models to section 4.3, such as attacking models trained by adversarial training ($\ell_{2}$ model with $\epsilon = 0.5$).

[Author Response · NeurIPS 2019]

We appreciate all reviewers for their helpful and constructive comments. We'll further improve the paper in the final
version. Below we address their detailed comments.

**R1: RGF outperforms NES:** The major difference between RGF and NES [16] is that NES adopts the antithetic
sampling, while RGF does not. Specifically, the gradient estimator is $\hat{g} = \frac{1}{q}\sum_{i=1}^{q}\frac{f(x+\sigma u_i,y)-f(x-\sigma u_i,y)}{2\sigma}u_i$ in NES and
$\hat{g} = \frac{1}{q}\sum_{i=1}^{q}\frac{f(x+\sigma u_i,y)-f(x,y)}{\sigma}u_i$ in RGF (see Eq.(5)). The NES estimator can eliminate the second-order component
of $f$ through central differences, but it requires $2q$ queries while RGF only requires $q+1$ queries. When $\sigma$ is small, the
second-order component is often dominated by the first-order one. So RGF outperforms NES. We'll make it clearer.

**R1: $\lambda^*$ distribution and cosine similar-**
**ity across the attack iterations:** Thanks
for the suggestion. As it's hard to plot
the full distribution of $\lambda^*$, which changes
during iteration, we show the average $\lambda^*$
over all images w.r.t. iterations in Fig. A.
It shows that $\lambda^*$ decreases along with the
iterations (i.e., the distribution concen-

Figure A: The average $\lambda^*$ across attack iterations.

Figure B: The cosine similarity across attack iterations.

Figure C: The estimation error with different $S$.

trates on small $\lambda^*$). Fig. B shows the cosine similarity between the transfer and the true gradients, and that between the
estimated and the true gradients, across iterations. The results show that the transfer gradient is useful at beginning, and
becomes less useful along with the iterations. However, the estimated gradient can remain higher cosine similarity with
the true gradient, which facilitates the adversarial attacks consequently. We'll add the results in the final version.

**R2: Novelty of the idea:** As stated in L108-113, we consider the score-based setting while [4] focuses on the decision-
based setting. [4] is built upon the Boundary method [3] and uses a fixed coefficient to incorporate the transfer gradient.
Due to the different settings, we introduce a new objective (see Eq.(7)) for gradient estimation, and optimize it inside
the proposed family of estimators, resulting in a generic P-RGF algorithm which incorporates the transfer gradient with
an optimal coefficient. Technically, it's non-trivial to derive the optimal solution. Moreover, we found that it's necessary
to use an adaptive coefficient rather than a fixed value since 1) the usefulness of the transfer gradient varies across
iterations; 2) experiments show that our algorithm is beneficial from the adaptive coefficient. Overall, we propose a
simple, yet novel and effective method, considering a different black-box setting from [4], as agreed by R1 and R3.

**R2: More analysis and experiments about the estima-**
**tion of gradient norm:** Thanks for the comment. The
gradient norm (or cosine similarity) is easier to estimate
than the true gradient since it's a scalar value. Fig. C shows

Table A: Additional experimental results.

| Methods | Inception-v3 | | VGG-16 | | ResNet-50 | |
|---|---|---|---|---|---|---|
| | ASR | AVG. Q | ASR | AVG. Q | ASR | AVG. Q |
| P-RGF ($\lambda = 0.05$) | 97.8% | 1021 | 99.7% | 888 | **99.6%** | 790 |
| P-RGF ($\lambda^*$, true norm) | **98.1%** | 768 | **99.8%** | **501** | 99.5% | **427** |
| P-RGF ($\lambda^*$) | **98.1%** | 745 | **99.8%** | 521 | **99.6%** | 452 |

the estimation error of the gradient norm, defined as the (normalized) RMSE—$\sqrt{\mathbb{E}\big(\frac{\|\widehat{\nabla f(x)}\|_2-\|\nabla f(x)\|_2}{\|\nabla f(x)\|_2}\big)^2}$, w.r.t. the
number of queries $S$. We chose $S = 10$ in all experiments to reduce the number of queries while the estimation error is
acceptable. We also show the overall attack results of using the true gradient norm instead of the estimated norm in
Table A (Row 2). The results are similar to those of using the estimated norm. We'll add the results in the final version.

**R2: Experiments about P-RGF with a fixed $\lambda = 0.05$:** Thanks for the suggestion. Table A (Row 1) shows the results
of P-RGF with $\lambda = 0.05$ (optimal in Fig. 1(b)), which are better than P-RGF with $\lambda = 0.5$ (in Table 1). However, a
significant performance gap still remains from using the adaptive $\lambda^*$. We'll add the results in the final version.

**R3: The improvement over the RGF method is not significant:** In Table A, P-RGF and RGF obtain similar attack
success rates. The reason is that the maximum number of queries (i.e., 10,000) is sufficient for them to find adversarial
perturbations, such that their attack success rates are similarly high. However, P-RGF requires fewer queries than RGF
($20\% \sim 45\%$ queries reduction). If the maximum number of queries is set to 1,000, the attack success rate against
Inception-v3 becomes $56.4\%$ using RGF and $78.6\%$ using P-RGF (the average number of queries is 470 and 297
respectively). Moreover, in Table 2, P-RGF obtains much higher success rates than RGF, and also reduces the query
complexity for attacking the defensive models. In summary, the improvement is significant in most of the cases.

**R3: Attack results on adversarially trained defensive models:** Thanks for the sug-
gestion. We choose [*1] as our target model, which successfully performs PGD-based
adversarial training on ImageNet. The gradient from ResNet-152 can hardly transfer to
this model, and the results of RGF and P-RGF are similar. So we use another adversarially
trained model (with a different architecture) to provide the transfer gradient. We perform $\ell_\infty$ attacks with $\epsilon = 16/255$,

Table B: Attack results on adversarially trained model.

| | ASR | AVG. Q |
|---|---|---|
| RGF | 31.7% | 1207 |
| P-RGF ($\lambda^*$) | **64.7%** | **378** |

which is the same threat model used in adversarial training. Table B presents the results—P-RGF outperforms RGF
significantly with the strong transfer-based prior. We'll add the results in the final version.

[*1] C. Xie, Y. Wu, L. van der Maaten, et al. Feature denoising for improving adversarial robustness. CVPR 2019.

[Meta-Review · NeurIPS 2019]

The paper proposes to use transfer-based priors to improve query-based black-box attacks. After checking the author response, all the reviewers agreed that this is a solid paper and should be accepted.